# The past, present and future impact of HIV prevention and control on HPV and cervical disease in Tanzania: A modelling study

**Michaela T. Hall**[1,2]*, **Megan A. Smith**[2,3], **Kate T. Simms**[2,3], **Ruanne V. Barnabas**[4], **Karen Canfell**[2,3], **John M. Murray**[1]

1 School of Mathematics and Statistics, UNSW Sydney, Kensington, Australia, 2 Cancer Research Division, Cancer Council NSW, Woolloomooloo, Australia, 3 School of Public Health, University of Sydney, Sydney, Australia, 4 University of Washington, Seattle, WA, United States of America

* michaela.hall@nswcc.org.au

**Data Availability Statement:** All relevant data are within the manuscript and its Supporting Information files.

## Abstract

### Background

Women with HIV have an elevated risk of HPV infection, and eventually, cervical cancer. Tanzania has a high burden of both HIV and cervical cancer, with an HIV prevalence of 5.5% in women in 2018, and a cervical cancer incidence rate among the highest globally, at 59.1 per 100,000 per year, and an estimated 9,772 cervical cancers diagnosed in 2018. We aimed to quantify the impact that interventions intended to control HIV have had and will have on cervical cancer in Tanzania over a period from 1995 to 2070.

### Methods

A deterministic transmission-dynamic compartment model of HIV and HPV infection and natural history was used to simulate the impact of voluntary medical male circumcision (VMMC), anti-retroviral therapy (ART), and targeted pre-exposure prophylaxis (PrEP) on cervical cancer incidence and mortality from 1995–2070.

### Findings

We estimate that VMMC has prevented 2,843 cervical cancer cases and 1,039 cervical cancer deaths from 1995–2020; by 2070 we predict that VMMC will have lowered cervical cancer incidence and mortality rates by 28% (55.11 cases per 100,000 women in 2070 without VMMC, compared to 39.93 with VMMC only) and 26% (37.31 deaths per 100,000 women in 2070 without VMMC compared to 27.72 with VMMC), respectively. We predict that ART will temporarily increase cervical cancer diagnoses and deaths, due to the removal of HIV death as a competing risk, but will ultimately further lower cervical cancer incidence and mortality rates by 7% (to 37.31 cases per 100,000 women in 2070) and 5% (to 26.44 deaths per 100,000 women in 2070), respectively, relative to a scenario with VMMC but no ART. A combination of ART and targeted PrEP use is anticipated to lower cervical cancer incidence and mortality rates to 35.82 and 25.35 cases and deaths, respectively, per 100,000 women in 2070.

**Funding:** This work was funded by an Australian Government Research Training Program (RTP) Scholarship (Ms Michaela Hall 5045590). The funders had no role in study design, data collection and analysis, decision to publish, or preparation of the manuscript.

**Competing interests:** KC is a co-PI of an investigator-initiated trial of primary HPV screening in Australia ('Compass') which is conducted and funded by the VCS Foundation, a government-funded health promotion charity, which has received a funding contribution from Roche Molecular Systems and Roche Tissue Diagnostics, AZ, USA. Neither KC nor her institution on her behalf have received funding from industry for this or any other project. This does not alter our adherence to PLOS ONE policies on sharing data and materials.

## Conclusions

HIV treatment and control measures in Tanzania will result in long-term reductions in cervical cancer incidence and mortality. Although, in the near term, the life-extending capability of ART will result in a temporary increase in cervical cancer rates, continued efforts towards HIV prevention will reduce cervical cancer incidence and mortality over the longer term. These findings are critical background to understanding the longer-term impact of achieving cervical cancer elimination targets in Tanzania.

## Background

For many years human immunodeficiency virus (HIV) has been one of the most heavily researched infectious diseases, and now, controlling HIV is beginning to look achievable [1]. Improved methods of HIV prevention and control such as pre-exposure prophylaxis (PrEP), anti-retroviral therapy (ART) and even voluntary medical male circumcision (VMMC) are at the forefront of health policy recommendations [1–4]. If these interventions are effectively implemented at the population level, they may substantially reduce HIV transmission and eventually end the HIV epidemic. Many modelling studies have attempted to quantify the impact of these interventions on HIV prevalence and related mortality in a range of settings [5–10].

HIV positivity has been linked to higher rates of human papillomavirus (HPV) acquisition, and, among those infected with HPV, the presence of an HIV co-infection is known to reduce the likelihood of HPV clearance and regression of pre-cancerous lesions, and, increase the risk of progression [11]. For this reason, modelling studies evaluating cervical cancer prevention policies are increasingly considering, either directly or indirectly, the impacts of endemic HIV in relevent settings [7, 12, 13].

Methods of HIV control may have a substantial impact not only on prevalence and deaths due to HIV, but also on HPV prevalence and subsequently cervical cancer incidence and mortality rates [11, 14]. In particular, male circumcision has been shown to reduce the risk of HIV-1 acquisition in heterosexual men over a time-period of 18–24 months by at least 60%, and, reduce HPV prevalence among heterosexual men by 63% [15–19]. Reductions in male HIV and HPV prevalence then results in women also experiencing less HIV and oncogenic HPV infection, and subsequently, less cervical cancer [16, 20]. A global ecological analysis classifying VMMC into high (>80%), intermediate (20–80%) and low (<20%) prevalence has reported that for each categorical shift in VMMC prevalence, cervical cancer incidence was reduced by 3.65 (0.54–6.76) cases per 100,000 women per year [21].

The United Republic of Tanzania has a high burden of both HIV and cervical cancer. It was estimated that in 2018, 5.5% of Tanzanian women aged 15–49 years were living with HIV [22], while the incidence of cervical cancer was among the highest globally, at 59.1 cases diagnosed per 100,000 women (9,771 cervical cancers detected) in 2018 [23]. The 2018 incidence rates of cervical cancer in Southern Africa and Eastern Africa were 43.1 cases per 100,000 women per year, and 40.1 cases per 100,000 women per year, respectively [23]. Tanzania is within the sub-Saharan African region, which in 2018 contained 53% of all people living with HIV globally and had an estimated HIV prevalence among adults aged 15–49 years of 7% [24, 25]. The two main interventions against HIV currently in place in Tanzania are ART and VMMC, which are both being actively scaled up [26]; while the Tanzanian Ministry of Health recommends PrEP use for those at significant risk of HIV acquisition, scale up of access to PrEP has been limited [27]. In light of Tanzania's high burden of cervical cancer and the known impact of

HIV, it is important to assess the impact of HIV control interventions that are currently being scaled up (ART and VMMC) or considered (PrEP) on not only HIV incidence and prevalence, but also rates of cervical cancer incidence and mortality. This exercise will provide important context to understanding the impact of scaling up prevention and treatment strategies to achieve cervical cancer elimination targets. Furthermore, while there exists significant variation in national laws pertaining to sexual identity and orientation, sex-work, and access to contraception across the African continent which affect local rates of sexually transmitted diseases [24], the relative impact of HIV interventions on cervical cancer incidence rates in Tanzania is likely to be broadly representative of the region.

The aim of this analysis, therefore, was to quantify the effect of HIV control actions to date on cervical cancer incidence and mortality in terms of rates, cancer diagnoses and lives saved, and to predict future cervical cancer incidence and mortality rates in Tanzania, in the context of scaled-up HIV control interventions.

## Methods

### Model overview and parameterisation

A detailed deterministic transmission-dynamic compartment model was developed to concurrently simulate the transmission and natural history of HIV, HPV 16/18, HPV 31/33/45/52/58 (referred to as HPV H5) and other oncogenic high-risk HPV types (referred to as HPV OHR) in the United Republic of Tanzania. While there are a range of transmission modalities for HIV and HPV, this platform simulates heterosexual transmission only (a simplifying assumption), as this is the dominant mode of transmission for both HIV and HPV in sub-Saharan Africa [28, 29]. The platform can simulate dual HIV and HPV infections, as well as infections with multiple HPV types, with and without ART. The model incorporates comprehensive demographic, sexual behaviour and natural history assumptions, and accounts for VMMC, ART and PrEP. The simulated population includes males, females and a separate subgroup of female commercial sex-workers (which females may be hired into or retire from), from ages 5 to 79 years, stratified by sex, five-year age group, sexual activity level, HIV and HPV infection, and treatment status. This model is comprised of 11,022,480 compartments, where simulated populations move between the states described in Table 1. Note that all persons in the simulated population are categorised by some combination of attributes: sex/career, age, sexual activity level, HIV infection status and natural history, HIV treatment status (note that no HIV negative individuals are treated with the exception of PReP which may be provided prophylactically for HIV prevention in HIV negative individuals), HPV 16/18 infection status and natural history, HPV H5 infection status and natural history, HPV OHR infection status and natural history and cervical cancer detection status and treatment. Note that only women can progress from HPV infections to cervical pre-cancer, and only women with cervical cancer may have cancer detected. The model implementation utilised for this analysis runs on a quarterly timestep (13 weeks).

The model's input parameters were specified primarily using empirical data; however, some parameters, particularly those unobservable or informed by survey data, were found through calibration using a trust region reflective algorithm. The parameter inputs found through calibration were the per-timestep volumes of sex-, age- and activity-group specific high-risk sexual contacts, the degree of age-assortative sexual mixing, annual fluctuations in population-level risk aversive behaviour, and the relative per-sex-act probability of HIV acquisition for females compared to males. These inputs were calibrated to estimated sex-specific HIV prevalence over time, as well as annual rates of new HIV infections obtained from

**Table 1. Model compartments exist for the cartesian product of sets (A) to (I).**

| SEX/ CAREER (A) | AGE (YEARS) (B) | SEXUAL ACTIVITY LEVEL (C) | HIV NATURAL HISTORY (D) | HIV ART STATUS (E) | HPV 16/18 NATURAL HISTORY (F) | HPV H5 NATURAL HISTORY (G) | HPV OHR NATURAL HISTORY (H) | CERVICAL CANCER DETECTION AND TREATMENT (I) |
|---|---|---|---|---|---|---|---|---|
| Male | 5–9 | General population sexual activity | Immune (PrEP) | Untreated | Immune (HPV vaccine) | Immune (HPV vaccine) | Immune (HPV vaccine) | No cervical cancer detected |
| Female | 10–14 | Elevated sexual activity | Susceptible | ART (no viral suppression) | Susceptible | Susceptible | Susceptible | Symptomatically detected localised cervical cancer |
| Female sex-worker | 15–19 | | Acute infection | ART (viral suppression) | HPV 16/18 infection | HPV 16/18 infection | HPV 16/18 infection | Symptomatically detected regional cervical cancer |
| | 20–24 | | Stage 1 (WHO clinical) | | CIN 1 | CIN 1 | CIN 1 | Symptomatically detected distant cervical cancer |
| | . . . | | Stage 2 (WHO clinical) | | CIN 2 | CIN 2 | CIN 2 | Screen detected localised cervical cancer |
| | 60–64 | | Stage 3 (WHO clinical) | | CIN 3 | CIN 3 | CIN 3 | Screen detected regional cervical cancer |
| | 65–69 | | Stage 4 / AIDS (WHO clinical) | | Undetected localised cervical cancer | Undetected localised cervical cancer | Undetected localised cervical cancer | Screen detected distant cervical cancer |
| | 70–74 | | | | Undetected regional cervical cancer | Undetected regional cervical cancer | Undetected regional cervical cancer | Cervical cancer survivor |
| | 75–79 | | | | Undetected distant cervical cancer | Undetected distant cervical cancer | Undetected distant cervical cancer | |

UNAIDS [22, 30, 31]. Note that different groupings in the presented age-range of calibration/ validation results are due to variation in reported age-ranges in the observed data.

**Demography.** Population demography encompasses compartments for sex/career (male, female, commercial sex-worker), age (five-year age-groups from 5–9 to 75–79 years) and sexual activity level (age- and sex-specific rates of propensity towards sexual risk-taking). The demography module accounts for population ageing, recruitment, natural mortality and assigns risk groups. The youngest simulated age group is 5–9 years, therefore recruitment represents the number of children born who survive to age five, and accounts for the age- and year-specific fertility rates of the simulated female population, as well as infant mortality. The per-timestep probability of any individual ageing to the next five-year age-group is calculated using the number of single-year ages in the age group, and the number of model iterations per year. For example, $\frac{1}{5}$ of individuals in the 10–14 year age-group turn 15 in any given year, and since there are four timesteps simulated per year, the probability of ageing from the 10–14 year group to the 15–19 year group is $\frac{1}{5} \times \frac{1}{4} = 0.05$. In calculating recruitment, annual fertility rates were sourced from the World Bank using the median fertility variant [32], whereas data on maternal age at birth was sourced from the United Republic of Tanzania Ministry of Finance and is based on the 2012 census [33]. The simulated population is subject to an age-specific probability of death resulting from any cause other than HIV or cervical cancer (other cause mortality). Age-and-year-specific mortality rates were derived using the projected year-on-year life tables reported by the United Nations Population Division, adjusted for HIV and cervical cancer mortality [34]. Finally, the demography module re-distributes the simulated population into two sex- and age-specific sexual activity groups (high-activity and general-activity)

and simulates the recruitment of women into a career of commercial sex work, and, their eventual retirement. The initial age-distribution was based on the 1960 Tanzanian population [35], with a sex ratio of 1 male to 1.03 females, based on data from the World Bank [36].

**Force of infection.** The model simulates HIV and HPV transmission between sexual partners, including interactions between commercial sex workers (CSWs) and their male clients. The population is compartmentalised into 'high activity' and 'general activity' sexual activity groups, which differ in their assumed number of sexual contacts per timestep. The number of sexual interactions per time-step implicitly accounts for new partners, the per-partnership frequency of sex, and relationship type (casual or monogamous). An age-dependent proportion of the female population were assumed to be CSWs, with an age-specific probability of seeking commercial sex defined for males. Furthermore, CSWs engage in both personal and commercial sexual interactions, with a pre-defined age-specific client volume per timestep.

The model platform calculates the sex- and age-specific per-timestep force of infection using age-specific partnership preferences, sexual activity group, HIV/HPV prevalence among sex partners, the per-sex-act probability of pathogen transmission (stratified by disease stage where applicable) and uptake of preventative interventions such as condom use (specified separately for commercial sex and general partnerships), VMMC prevalence and ART use. For example, the force of infection for HIV for a male aged $a$ in sexual activity group $r$ at time $t$ is calculated using Eq 1.

$$\Lambda_M^{HIV}(a,r,t) = c_M(a,r,t)(1-\kappa_{HIV}(t))(1-\upsilon_{HIV}(t))\sum_i\left(\sum_{Tx}(\rho_M(a)\cdot T_{FM}^{HIV(Tx)}(i))\,I_F^{HIV(Tx)}(i,t)\right)+\lambda_M^{HIV}(a,r,t)\;\#(1)$$

$c_M(a,r,t)$ denotes the average number of sexual contacts for a male aged $a$ in sexual-activity group $r$ at time $t$; $k_{HIV}(t)$ denotes the per-sex-act probability that a condom is worn and prevents HIV acquisition; $\upsilon_{HIV}(t)$ denotes the probability that the male has undergone VMMC and the per-sex act-probability that this prevents HIV acquisition; $i$ is the stage of HIV disease among female sex-partners; $\rho_M(a)$ is a vector of the distribution of preferences for female partners of each age-group for males aged $a$ (vector over all ages summing to unity); $T_{FM}^{HIV(Tx)}(i)$ is the HIV-stage-specific per sex-act female-to-male transmission probability for females with treatment status $Tx$; $(a)\cdot T_{FM}^{HIV(Tx)}$ refers to the dot product between vectors $\rho_M(a)$ and $T_{FM}^{HIV(Tx)}$; $I_F^{HIV(Tx)}(i,t)$ is a vector specifying the age-specific probability of a female being HIV positive (simulated), stage $i$, and with treatment status $Tx$ and time $t$; and finally, $\lambda_M^{HIV}(a,r,t)$ is the probability of acquiring an HIV infection from a commercial sex worker. Note that

$$\lambda_M^{HIV}(a,r,t) = \varsigma_M(a,r,t)(1-\kappa_{HIV}(t))(1-\upsilon_{HIV}(t))\sum_i\left(\sum_{Tx}(\rho_M(a)\cdot T_{FM}^{HIV(Tx)}(i))\,I_{CSW}^{HIV(Tx)}(i,t)\right)\;\#(2)$$

Where $\varsigma_M(a,r,t)$ denotes the average number of commercial sexual contacts assumed for a male aged $a$ in sexual risk group $r$ at time $t$; and, $I_{CSW}^{HIV(Tx)}(i,t)$ is a vector specifying the age-specific probability of a commercial sex worker being HIV positive, stage $i$ and with treatment status $Tx$ at time $t$.

The HPV 16/18 force of infection for a high activity male aged $a$ at time $t$ is calculated in a similar way but simplified by the assumption that the probability of HPV transmission is fixed irrespective of HPV disease stage. That is,

$$\Lambda_M^{HPV_{1618}}(a,r,t) = c_M(a,r,t)(1-\kappa_{HPV}(t))(1-\upsilon_{HPV}(t))T_{FM}^{HPV_{1618}}\left(\rho_M(a)\cdot I_F^{HPV_{1618}}(t)\right)+\lambda_M^{HPV_{1618}}(a,r,t)\#(3)$$

$k_{HPV}(t)$ denotes the per-sex-act probability that a condom is worn and prevents HPV acquisition; $\upsilon_{HPV}(t)$ denotes the probability that the male has undergone VMMC and the per-sex act-probability that this prevents HPV acquisition; $T_{FM}^{HPV_{1618}}$ is the per sex-act female-to-male

HPV16/18 transmission probability; and, $I_F^{HPV_{1618}}(t)$ is a vector specifying the age-specific probability of a female being HPV 16/18 positive. Additionally,

$$\lambda_M^{HPV_{1618}}(a, r, t) = \varsigma_M(a, r, t)(1 - \kappa_{HPV}(t))(1 - \upsilon_{HPV}(t))T_{FM}^{HPV_{1618}}(\rho_M(a) \cdot I_{CSW}^{HPV_{1618}}(t)) \qquad (4)$$

Equations specifying the force of infection for HPV H5 and HPV OHR in males and females are similar to the above and are provided in S1 File. Detailed input parameter assumptions relevant to the calculation of force of infection are also described in S1 File (see equations s1-s12).

**Disease natural history.** Disease progression for HIV infection is governed by the following state diagram (Fig 1), where specific progression rates are dependent on the current stage of disease and treatment status.

The natural history of HIV infection progresses from acute HIV infection though four clinical disease stages. These stages are aligned with the World Health Organisation (WHO) Clinical Staging of HIV/AIDS for Adults and Adolescents [37], and are defined in terms of patient symptoms. Input parameters specifying HIV progression rates in the model are described in Table 2.

The model platform accounts for the detailed and well-understood natural history of HPV. Disease progression and regression for HPV infection are governed as per Fig 2, where specific progression rates are dependent on HPV type, age, disease stage, HIV positivity and ART treatment status. HPV types 16/18 are known to be more aggressive than other oncogenic HPV types, with elevated disease progression rates and reduced regression rates. Women with an HIV co-infection also experience more aggressive HPV infections; however, viral suppression through ART can help to mitigate this [11]. The model contains interacting compartments for all HPV susceptibility/infection and natural history states for HPV types 16/18, HPV H5 and HPV OHR. These stages are described in Table 3, and their interactions are summarised in Fig 2, which describes the state transitions possible from each state at the start of a new timestep, including the case where no state transition is made.

The model explicitly simulates the natural history of human papillomavirus infection for HPV types 16/18, HPV H5 and HPV OHR. The stage-, age- and HPV type-specific progression and regression rates are as published in previous analyses [39], and have been reproduced in S1 File. HIV positivity status and viral suppression through ART both impact HPV

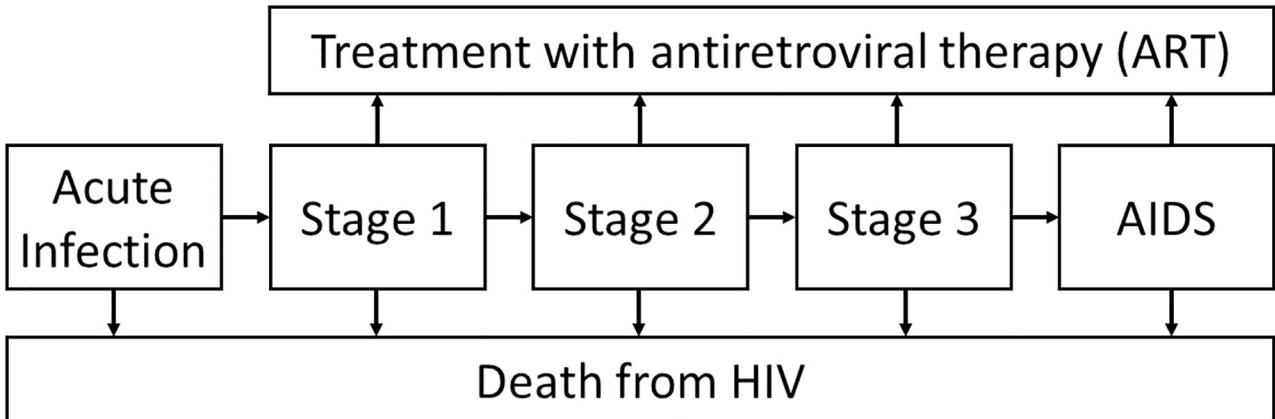

**Fig 1. State-space diagram for HIV disease progression.** Note that viral suppression was assumed to halt disease progression and that all states are subject to other cause mortality.

**Table 2. Average length of time spent in each disease stage, and the probability of HIV-death for each HIV disease stage.**

|  | Acute infection | WHO clinical stage 1 | WHO clinical stage 2 | WHO clinical stage 3 | WHO clinical stage 4 (AIDS) |
|---|---|---|---|---|---|
| Average length of time spent in HIV disease stage[1] | < 3 months | 1 year 6 months | 1 year 3 months | 6 years 6 months | 1 year 3 months |
| Per-timestep probability of HIV-death while in each disease stage[2] for ages 15–49 years and ages 50+, respectively | 3%, 5% | 2%, 5% | 4%, 4% | 1%, 2% | 14%, 27% |

[1]. As published in Palk et al 2018 (Sci Rep) [38];

[2]. As published in Tan et al [11].

acquisition and natural history; assumptions regarding the impact of HIV positivity on HPV natural history are summarised in S1 File.

**Interventions.** The model accounts for the impact of a range of HIV control interventions, including uptake of ART, PrEP, VMMC, and behavioural factors including the use of condoms. Effective use of ART in an individual infected with HIV not only acts to reduce disease progression and HIV-death but also significantly reduces the infectivity of virally suppressed patients. Further, use of PrEP, VMMC and condoms all lower the probability of disease acquisition to varying degrees. In the model, VMMC is specified by year, and we assumed rates were as reported in the literature and Tanzanian DHS reports [40–42]. The modelled VMMC rate applies to males of all ages, and reduces female to male HIV transmission by 60%, as consistent with the available evidence [43].

ART is also considered in two categories: those who are receiving ART, and those who are receiving ART and are 'virally suppressed'. The model assumes some mortality benefit for all individuals receiving ART, with viral suppression completely halting disease progression and/ or HIV death, and reducing infectiousness by 96% [13]. The percentage of virally suppressed people living with HIV (PLHIV) in Tanzania was assumed to match figures published by UNAIDS [44].

## Scenarios and outcomes

A range of counterfactual and potential future HIV epidemic control scenarios were simulated, as described in Table 4. For each scenario, we estimated cervical cancer incidence and cervical cancer mortality (stratified by HIV positivity) from 1995–2020, projecting these outcomes based on model hypotheses from 2020–2070. The absolute numbers of cervical cancer cases and deaths prevented by interventions to date (2020) are presented in this analysis, in addition to an age-standardised rate (ASR). The age-standardised rate is a weighted mean of the age-specific rates where weights (summing to unity) are derived from the 2015 estimated world female population [35], presented per 100,000 women.

## Sensitivity analysis

A multivariate sensitivity analysis was carried out to assess the robustness of model outcomes to variation in a range of parameters. Parameters were selected for sensitivity analysis if they were either difficult to observe/report on, suspected to be highly influential, or directly affect the interventions assessed in the scenario analysis. The modelled effect of HIV control interventions is dependent on assumptions about the magnitude of their effectiveness, and, the literature indicates uncertainty surrounding the effect of VMMC, ART and PrEP [11, 16, 46–48]. Furthermore, any population or individual level behavioural change driven by implementation of these interventions is difficult to quantify, as perceptions of risk are constantly changing. However, evidence suggests that the availability and uptake of HIV control interventions

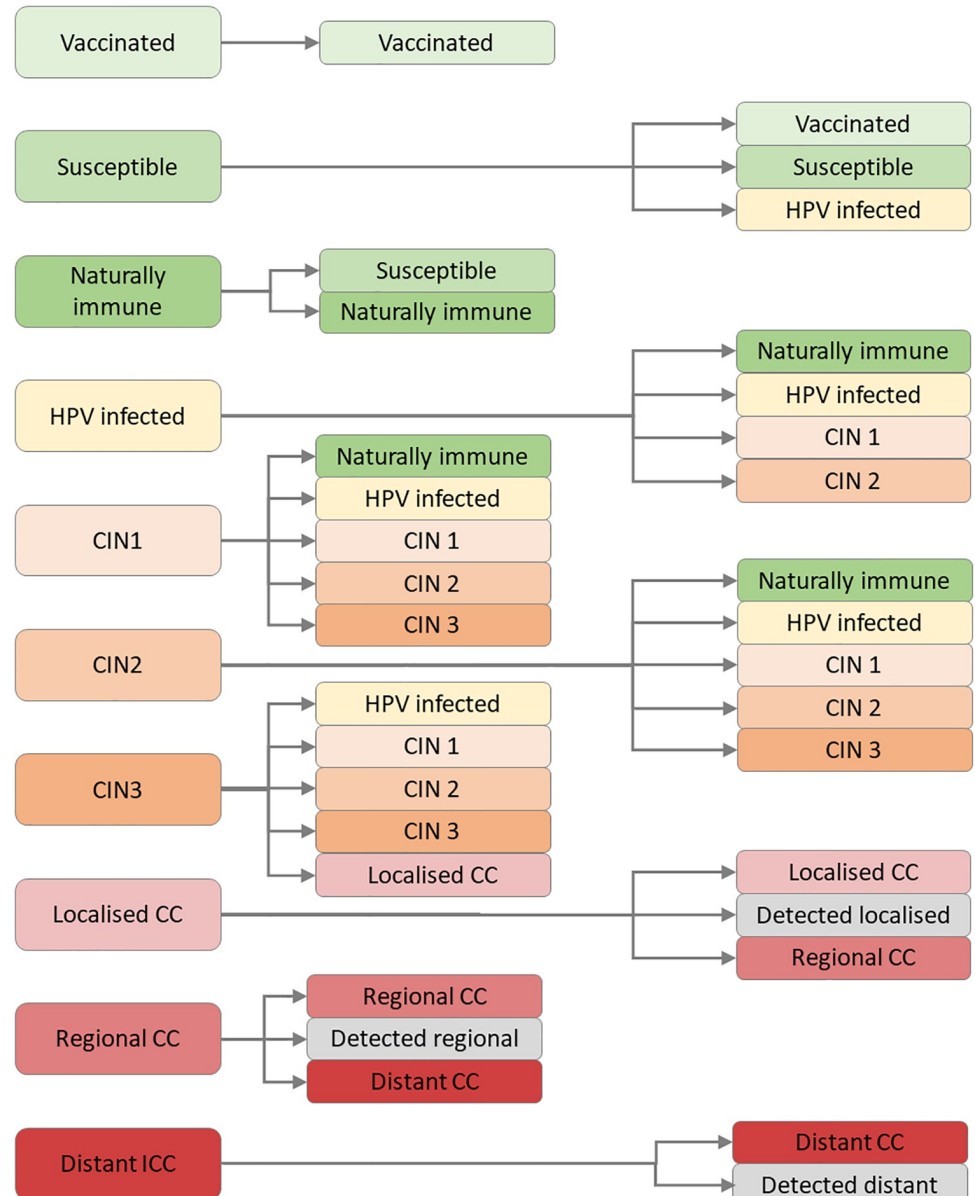

**Fig 2. State space diagram for the natural history of HPV and cervical cancer carcinogenesis; note that all compartments are subject to natural mortality, and detected cancer (grey) compartments are subject to stage-specific cervical cancer mortality and survival rates; CIN = cervical intraepithelial neoplasia; CC = cervical cancer.**

may facilitate an increase in risky sexual practices to the order of up to 21% [49–54]. A Latin Hypercube Sampling (LHS) analysis over 6,000 possible parameter sets was utilised, the values of which are described in Table 5.

# Results

## Calibration and validation

Simulations from the calibrated model were consistent with observed HIV-specific outcomes including male and female HIV prevalence, total HIV incidence and number of HIV deaths (Fig 3), in addition to age-specific 2018 cervical cancer incidence and mortality rates (Fig 4).

**Table 3. List and description of HPV transmission and natural history compartments.**

| Compartment name | Description |
| --- | --- |
| Vaccinated | Vaccinated individuals are unable to become infected with HPV. Note that this model iteration does not assume any HPV vaccination occurs. |
| Susceptible | Individuals are recruited into this compartment. |
| Naturally immune | Once clearing an HPV infection, some individuals retain temporary natural immunity. |
| HPV infected | Susceptible individuals may become infected with HPV. Initial model conditions specify a small number of HPV infected individuals to start the simulation. |
| CIN 1 | Cervical intraepithelial neoplasia (abbreviated as CIN) stage 1 is a low-grade pre-cancerous lesion. |
| CIN 2 | CIN stage 2 is a high-grade pre-cancerous lesion. |
| CIN 3 | CIN stage 3 is a high-grade pre-cancerous lesion. |
| Localised cervical cancer (undetected) | Localised cervical cancer (undetected) is the early-stage cervical cancer state. |
| Localised cervical cancer (detected) | Women with detected cancer (any stage) are not explicitly subject to disease progression or regression. Women with detected cancer (any stage) are subjected to a probability of cervical cancer mortality, increases with stage and which removes them from the model; alternatively, they may survive their disease. |
| Regional cervical cancer (undetected) | Regional cervical cancer (undetected) is the mid-stage cervical cancer state. |
| Regional cervical cancer (detected) | See localised cervical cancer (detected). |
| Distant cervical cancer (undetected) | Distant cervical cancer (undetected) is the late-stage cervical cancer state (the cancer has metastasised and has low survival probability). |
| Distant cervical cancer (detected) | See localised cervical cancer (detected). |
| Cervical cancer survival | Detected cervical cancer is in remission. Individuals in this compartment are no longer subject to cervical cancer mortality. |

Following the model calibration procedure, where parameters were chosen such that the model was a good fit to UNAIDS and Globocan (IARC) data [22, 23, 30, 31], the model was validated against independent datasets. These included sex- and age-specific HIV prevalence (Fig 5A and 5B) and the sex-specific age distribution of AIDS diagnoses (Fig 5C and 5D), age-specific HPV prevalence (Fig 6), and the prevalence of HSIL (high-grade squamous intra-epithelial lesion; considered equivalent to a diagnosed CIN 2/3) among HIV negative versus positive women from the PROTECT study [58] (Fig 7).

## Scenario analysis

The model estimates that there were 2,843 (and 1,039) fewer cervical cancer cases (and deaths), and 33,648 fewer total deaths (HIV and cervical cancer combined) from 1995 to 2020 as a result of the introduction and scale-up of VMMC. Assuming VMMC is maintained at 2018 levels, by 2070, VMMC is expected to have averted 330,400 (and 186,260) cervical cancer cases (and deaths) and save 3.47 million lives (HIV and cervical cancer combined) (Table 6; Fig 8B and 8D). ART use to 2020 is estimated to have led to an additional 1,573 (and 1,222) cervical cancer cases (and deaths) from 1995–2020; this is due to ART reducing the competing risk of mortality due to HIV. The cumulative number of deaths averted by ART is predicted to reach 2,253,700 by 2070 and will have prevented 52,430 (and 16,390) additional cervical cancer cases (and deaths) by 2070 (Table 6; Fig 8B and 8D).

VMMC is predicted to substantially reduce both HPV and HIV prevalence in Tanzanian men and women (Table 7, Fig 9). By 2070, VMMC is predicted to reduce HPV prevalence in

**Table 4. Modelled scenarios of HIV control.**

| Scenario name and description | VMMC assumptions $v(t)$[1] | ART assumptions[2] | PrEP assumptions |
|---|---|---|---|
| **No interventions**. Exploratory worst-case counterfactual scenario. | No VMMCs carried out. | No ART uptake. | No PrEP available. |
| **VMMC only**. Exploratory pessimistic counterfactual scenario. | VMMC as historically observed, maintained at 80% from 2018 onwards. | No ART uptake. | No PrEP available. |
| **VMMC and ART (baseline)**. Baseline scenario reflecting the current situation in Tanzania. | VMMC as historically observed, maintained at 80% from 2018 onwards. | Proportion of PLHIV receiving ART and virally suppressed as historically observed, and maintained at 47% from 2018 onwards. | No PrEP available. |
| **VMMC and target ART**. Exploratory optimistic scenario. | VMMC as historically observed, maintained at 80% from 2018 onwards. | Proportion of PL HIV receiving ART and virally suppressed as historically observed, and scaled up from 47% in 2018 to meet WHO '90-90-90' HIV control targets[3] from 2020. | No PrEP available. |
| **VMMC, target ART and PrEP**. Exploratory best-case scenario. | VMMC as historically observed, maintained at 80% from 2018 onwards. | Proportion of PLHIV receiving ART and virally suppressed as historically observed, and is scaled up from 47% in 2018 to meet WHO '90-90-90' HIV control targets from 2020. | Daily PrEP use available to women engaging in commercial or transactional sex and their clients (90% uptake, 99% efficacy) from 2020. |

[1]VMMC as historically observed assumes 8% VMMC until 1995, increasing to 23% in 1998, then increases quadratically to 80% in 2015, as observed [40, 42].

[2]ART as historically observed assumes introduction of ART in 2005, with a linear scale-up to 47% virally suppressed in 2017 [44].

[3]The 90-90-90 targets refer to the WHO global goal of achieving the following: 90% of all people living with HIV aware of their HIV status, 90% of all people diagnosed with HIV receive sustained ART, and 90% of all people receiving ART achieve viral suppression [45].

men and women by 28% (44.71% under 'No interventions' to 32.13% under 'VMMC only' in 2070) and 17% (46.39% under 'No interventions' to 38.56% under 'VMMC only'), respectively. Similarly, the estimated reduction in HIV prevalence due to VMMC is 75% (7.59% under 'No interventions' to 1.86% under 'VMMC only') and 71% (8.47% under 'No interventions' to 2.46% under 'VMMC only'), respectively, for men and women. Compared to the 'No intervention' scenario, the introduction and scale-up of ART and PrEP is estimated to reduce HIV prevalence in men and women by approximately 99% (to 0.05% in males and 0.11% in females) in 2070.

The introduction and scale-up of HIV preventions is expected to reduce the age-standardised rates of cervical cancer incidence and mortality over time. VMMC is expected to reduce cervical cancer incidence and mortality rates by 36–40% in 2070 compared to 2020 rates (under 'VMMC and ART (baseline)' scenario), whereas the provision and scale-up of ART to meet World Health Organisation 90-90-90 HIV control targets reduces cervical cancer incidence and mortality rates by 41–45% in 2070 compared to the current rates in 2020. Absolute rates of cervical cancer incidence and mortality are presented in Table 8 and visualised in Fig 10. Here, we note that the reductions in cervical cancer incidence and mortality due to ART among all women are driven by reductions among HIV positive women.

## Sensitivity analysis

Findings from the multivariate sensitivity analysis indicate that the simulation outcomes are highly sensitive to variation in parameters specifying sexual behaviour, disease transmission and natural history, and, intervention effectiveness. Fig 11 summarises the baseline and total variation in 2070 endpoint predictions for all simulated outcomes over the five scenarios.

An analysis of partial rank correlation coefficients indicate that cervical cancer incidence is most strongly correlated with VMMC efficacy for HPV prevention (correlation coefficient of

**Table 5. Parameter variation considered in LHS analysis, and the rationale for selecting these parameters/ranges.** Parameters are described in S1 Table in S1 File.

| Parameter | Value ranges in sensitivity analysis | Rationale |
|---|---|---|
| **Intervention and behavioural parameters** | | |
| **Sexual behaviour.** Per-timestep volume of sexual interactions possibly resulting in HIV transmission for high- and general activity males and females. The parameters varied were $c_M(a, r, t)$ and $c_F(a, r, t)$. | Parameters were varied by ±5% of the baseline value. The baseline values for these parameters are specified explicitly in S1 File. | This variation assessed the model's sensitivity to these parameters, while not producing excessive variation in simulation outcomes. |
| **Condom usage.** Population-level behavioural change (i.e. disinhibition) resulting from uptake/availability of HIV control. The parameters varied were $k_{HIV}(t)$ and $k_{HPV}(t)$. | Condom usage from 1990 to 2016 is reported in S1 Table in S1 File and reflects observed usage in Tanzania, with the assumption that a condom was used in 37% of all high-risk interactions from 2016 onwards. In sensitivity analysis we considered variation in condom use from 2020 onwards, and the percentage of all high-risk interactions where a condom is used was varied uniformly over the interval 32–42%. | A range of studies reported that availability of HIV control interventions ART and PrEP can facilitate behavioural disinhibition where risk behaviours are increased in a population [49–54]. |
| **Mixing dimension.** Age-assortative mixing. The parameter varied was $\lambda_{max}$. | The maximum Poisson parameter $\lambda_{max}$ was varied over a range $\lambda_{max} = 1.5$ (1.4, 1.6). | Observed data indicates that males tend to mate with females younger than themselves [55]. |
| **VMMC efficacy (HIV).** Per sex-act reduction in HIV acquisition risk for circumcised males (compared to uncircumcised males). The parameter varied was $v_{HIV}(t)$. | Relative risk reduction assumed in sensitivity analysis: 0.53–0.6 (0.6 assumed at baseline). | Observed data indicates that circumcised men are 60% (53%-60%) less likely to acquire an HIV infection than uncircumcised men [46]. |
| **VMMC efficacy (HPV).** Per sex-act reduction in HPV acquisition risk for circumcised males (compared to uncircumcised males). The parameter varied was $v_{HPV}(t)$. | Relative risk reduction assumed in sensitivity analysis: 0.16–0.85 (0.63 assumed at baseline). | A pooled analysis reported that circumcised men are 37% (95% CI: 16%-85%) less likely to have an HPV infection than uncircumcised men [16]. |
| **HPV natural history on ART.** Impact of viral suppression through ART on HPV persistence and natural history among HIV infected women. | Baseline assumption: ART reduces additional risk of HPV/pre-cancer progression due to HIV positivity by 50%. Sensitivity analysis: ART reduces additional risk of HIV/pre-cancer progression by 0–100%.<br><br>The specific values of multipliers specifying progression/regression of HPV related disease in HIV positive women compared to HIV negative women are outlined in S4 Table in S1 File. | A study, comparing relative risk of HPV acquisition among virally suppressed women to those not receiving ART, found that ART decreased HPV incidence (OR 0.64; 95% CI 0.46–0.88), but that this was not the case for high-risk HPV (OR 0.62; 95% CI 0.38–1.02) [47]. Another study found that women receiving ART are at lower LSIL risk (RR 0.67; 95% CI 0.45–1), and were 2.29 times more likely to regress from LSIL (95% CI 1.56–3.37) [11]. Overall, a systematic review and meta-analysis by Liu et al reported that the impact of ART on HPV-related disease is unclear [11]. |
| **PrEP efficacy.** Per sex-act reduction in HIV acquisition for daily PrEP users compared to non-PrEP users. | Baseline relative risk reduction: 0.99. Range assumed in sensitivity analysis: 0.92–0.99. | The iPrEx study found that PrEP can reduce HIV risk by 92%-99% for daily use among HIV-negative individuals [48]. |
| **Underlying transmission and natural history parameters** | | |
| **HIV transmission.** Stage-specific per-contact probability of HIV transmission | Multipliers against the base HIV transmission probability: Multiplier for 'WHO clinical stage 2' ($p_2$) = 4.3 (2.25, 17.91) | Ranges are derived from confidence intervals given in Quinn et al. [56]. |
| | Multiplier for 'WHO clinical stage 3' ($p_3$) = 6.5 (2.93, 19.97) | |
| | Multiplier for 'WHO clinical stage 4' ($p_4$) = 8.7 (5.28, 36.99) | |
| **HIV acquisition (female-to-male ratio).** The relative risk of HIV acquisition per sexual contact for females compared to males. The parameter varied was $p_f$ where $T_{MF}^{HIV(Tx)} = p_f T_{FM}^{HIV(Tx)}$. | The multiplier $p_f$ was found to be 2.5 through calibration and was varied between 2.2 and 2.8 in sensitivity analysis. | Ranges were based on observed data published in Quinn et al. [56]. |
| **HPV transmission.** Type-specific HPV transmission probabilities per sex act. | $T_{FM}^{HPV_{1618}} = 0.056 - 0.078$ | This variation assessed the model's sensitivity to these parameters, while not producing excessive variation in simulation outcomes. |
| | $T_{FM}^{HPV_{H5}} = 0.0134 - 0.0448$ | |
| | $T_{FM}^{HPV_{OHR}} = 0.0202 - 0.056$ | |
| | Where $T_{FM}^{HPV} = T_{MF}^{HPV}$ for all HPV types. | |

*(Continued)*

**Table 5.** (Continued)

| Parameter | Value ranges in sensitivity analysis | Rationale |
|---|---|---|
| **HIV-dependent HPV natural history.** HPV-type specific multipliers for acquisition, progression and regression of HPV associated disease for HIV positive individuals. | The ranges considered are as described in in S4 Table in S1 File. | Ranges were based on data published in Liu et al 2018 [11]. |

-0.25; Fig 12), followed by VMMC efficacy for HIV prevention (correlation coefficient of -0.13).

## Discussion

To our knowledge, this analysis is the first to directly estimate the impact over time of changing VMMC prevalence, ART utilisation and PrEP uptake on cervical cancer in any setting. Findings from this analysis are likely to be broadly applicable to other low-income settings with high HIV prevalence and cervical cancer incidence rates, particularly in sub-Saharan Africa. These HIV control interventions were found to have a substantial impact on cervical cancer incidence and mortality in Tanzania. Using a simulation model of HIV and HPV

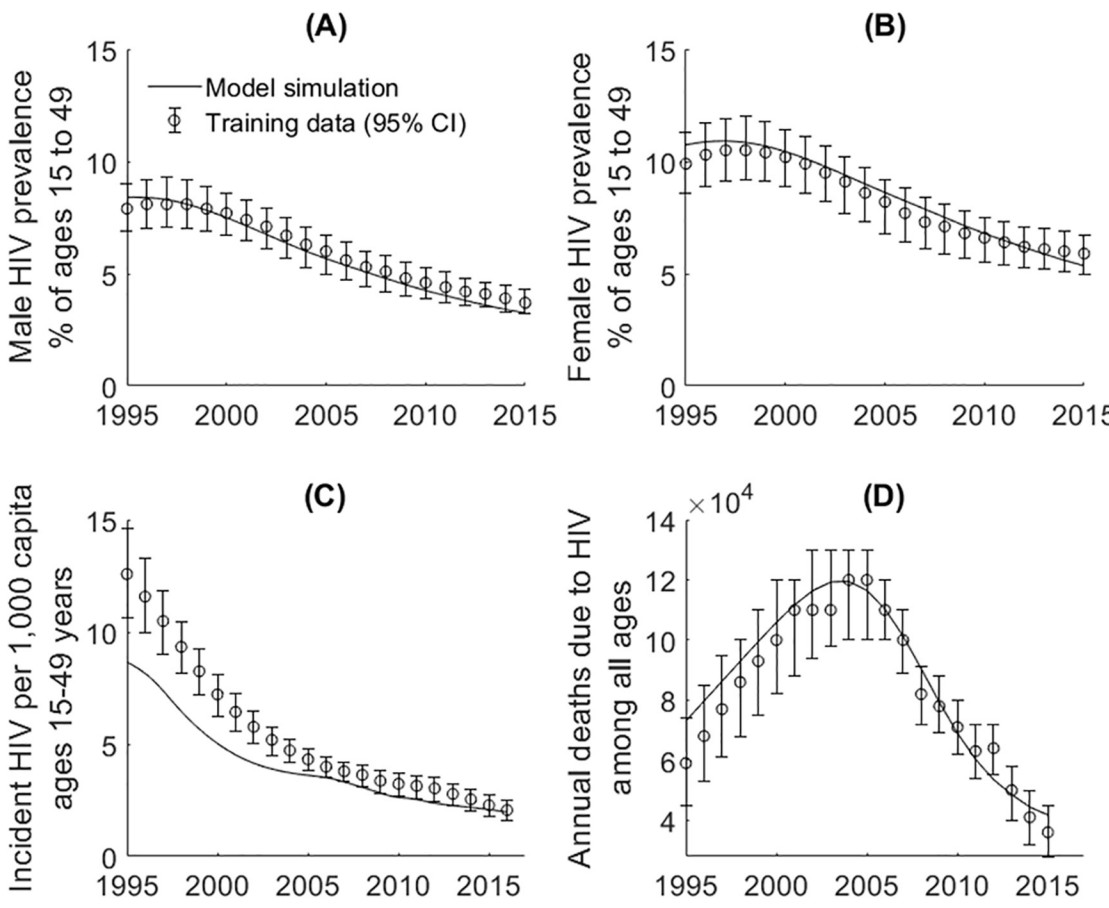

**Fig 3. Calibrated HIV outcomes.** (A) and (B) male and female HIV prevalence from 1995 to 2015; (C) HIV incidence from 1995 to 2015; (D) number of HIV deaths from 1995 to 2015. Error bars are 95% CI of observed data. Training data sourced from UNAIDS [22, 30,31,57].

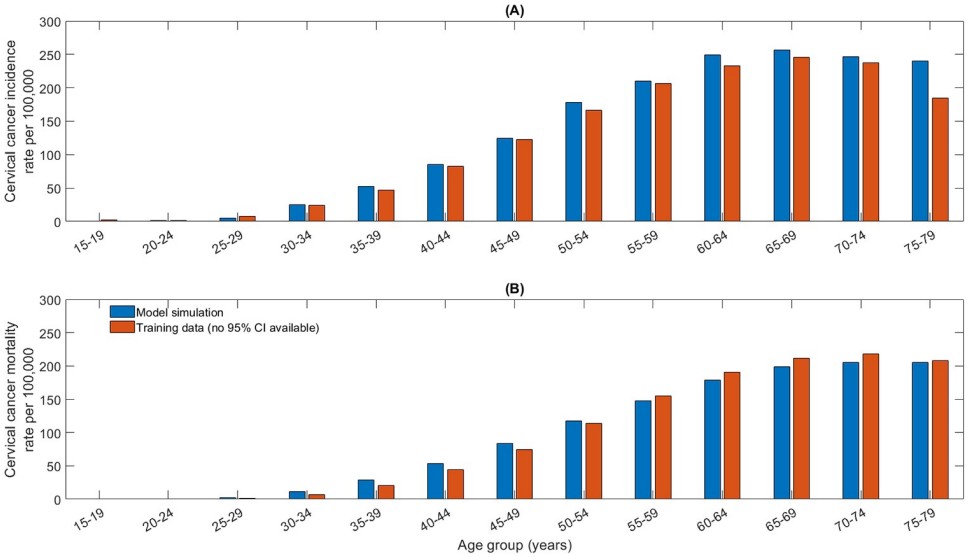

**Fig 4. Calibrated (A) age-specific cervical cancer incidence and (B) mortality for the year 2018 compared to estimated data sourced from the International Agency for Research on Cancer IARC [23].**

transmission to estimate cervical cancer cases, cervical cancer deaths and total deaths (including HIV deaths) in Tanzania from 1995 to 2070 in the context of currently implemented HIV control measures, we estimated that VMMC has prevented 2,843 cervical cancer cases and 1,039 cervical cancer deaths from 1995 to 2020. Perhaps a less intuitive finding is that, while the addition and scale-up of ART in HIV-positive women reduces both overall HIV and HPV prevalence (women effectively treated with ART are less likely to acquire HPV and more likely to clear an HPV infection than their untreated counterparts), ART is estimated to have

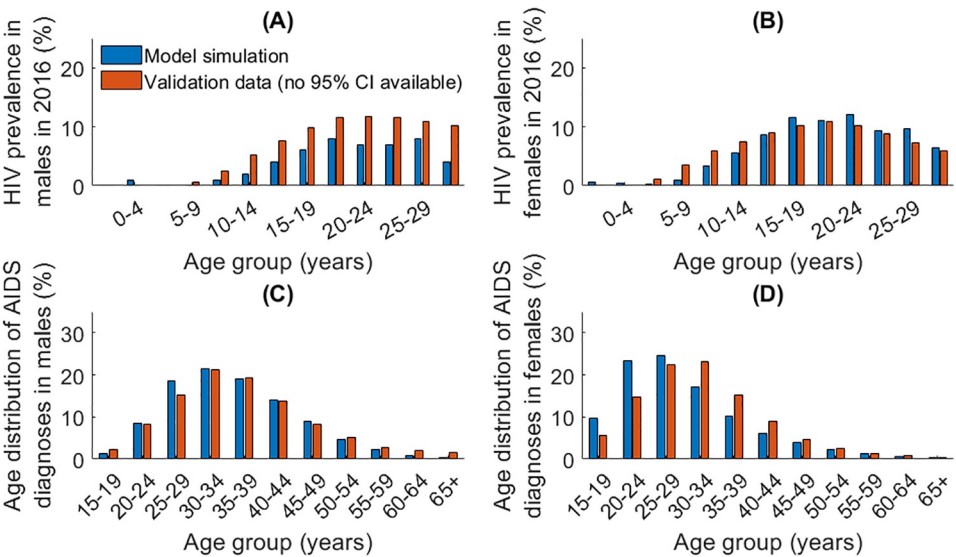

**Fig 5. (A) and (B) Age-specific HIV prevalence for males and females in 2016 compared to observed data; (C) and (D: age-distribution of AIDS diagnoses for males and females in 2011 compared to observed data.** Observed data from the Tanzanian Ministry of Finance (no confidence intervals available) [26, 59].

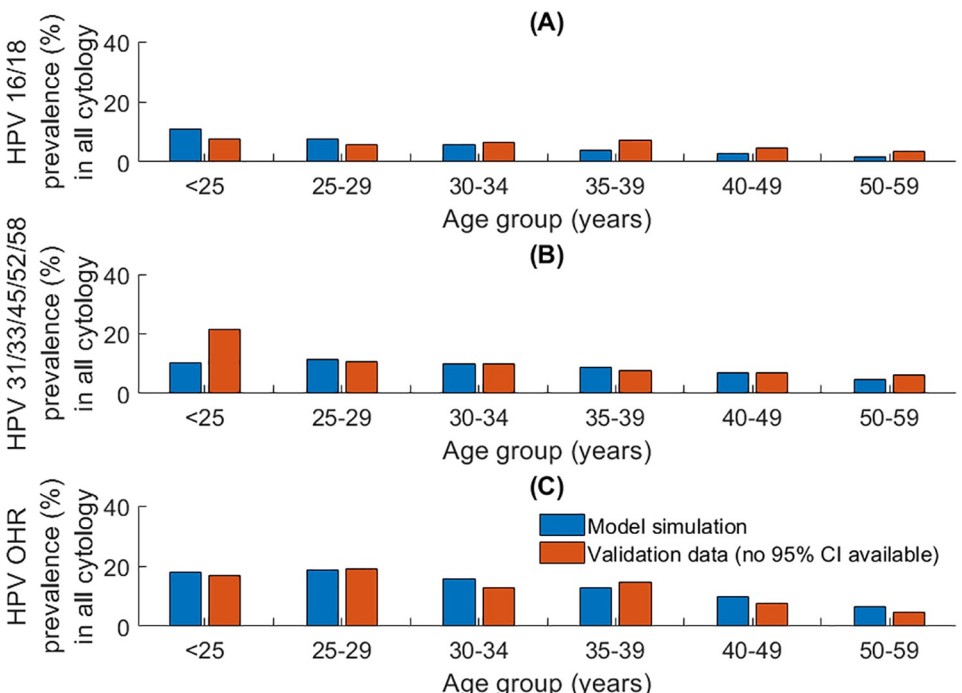

**Fig 6. (A), (B) and (C) Age-specific HPV prevalence in cervical cytology (all cytological results) of HPV 16/18, HPV H5 and HPV OHR compared to observed data.** Observed data from Dartell et al 2012 (no confidence intervals available) [58].

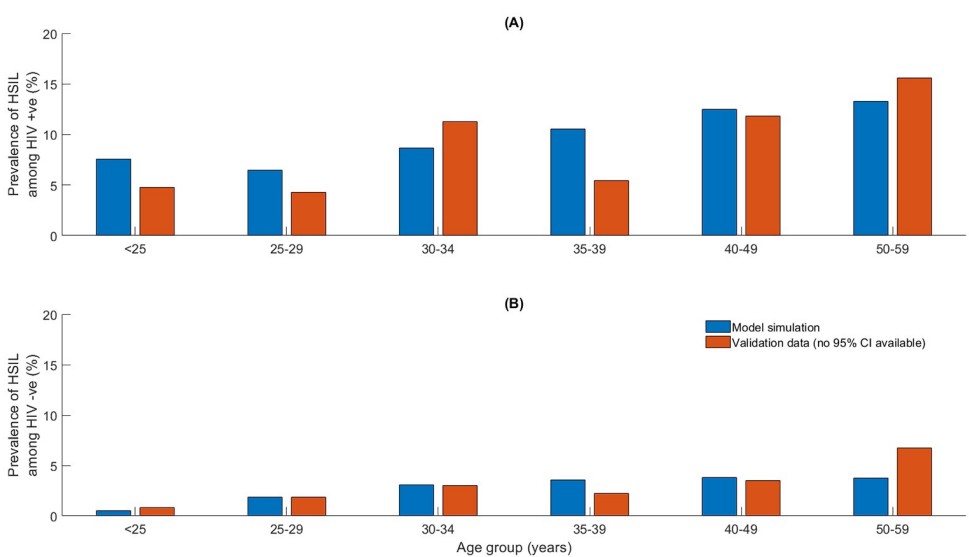

**Fig 7. Age specific rates of HSIL (detected high-grade squamous intraepithelial lesion consistent with CIN2/3) prevalence for (A) HIV positive and (B) HIV negative women.** Observed data from Dartell et al 2012 (no confidence intervals available) [58].

**Table 6. Number of prevented cervical cancer cases (and deaths) due to VMMC and ART.** Note that values presented reflect the incremental benefit of each intervention, and that negative values denote additional cases/deaths.

|  | Prevented by VMMC | Prevented by ART |
|---|---|---|
| Cervical cancer cases (deaths) prevented cumulatively from 1995 to 2020 | 2,843 (1,039) | -1,573 (-1,221) |
| Deaths due to HIV or cervical cancer (total deaths) prevented cumulatively from 1995 to 2020 | 33,648 | 147,262 |
| Cervical cancer cases (deaths) prevented cumulatively from 1995 to 2070 | 330,400 (186,260) | 52,430 (16,390) |
| Deaths due to HIV or cervical cancer prevented cumulatively from 1995 to 2070 | 3,469,800 | 2,253,700 |

resulted in some 1,573 additional cervical cancer diagnoses and 1,221 additional cervical cancer deaths, cumulatively, from 1995 to 2020. These additional cases and deaths are among HIV infected women and are caused by the removal of HIV-related death as a competing risk, as some women who would have otherwise died from HIV-related causes will develop cervical cancer and subsequently die from it, in the absence of scaled-up cervical cancer prevention. In the longer-term, the protective effect of ART prevails, as scale-up to meet World Health Organisation 90-90-90 HIV control targets would result in cervical cancer incidence and mortality rates that are 43% (37.31 c.f. 65.42 cases per 100,000 women per year) and 39% (26.44 c.f. 43.27 deaths per 100,000 women per year) lower, respectively, in 2070 compared to the current rates in 2020.

The model prediction for HIV prevalence over time is consistent with empirical data for Tanzania [22, 30], and, future predictions are broadly consistent with the findings from HIV modelling studies specific to sub-saharan Africa [60]. A comparative modelling study utilising predictions from four independent models predicts that under a scenario assuming universal HIV testing and treatment (up to 90% coverage), HIV prevalance will be reduced to 0–3% in sub-Saharan Africa in 2050. The current analysis predicts that HIV prevalence in Tanzania in 2050 will be 0.12% in males, and 0.19% in females aged 15–49 years in the 'VMMC and target

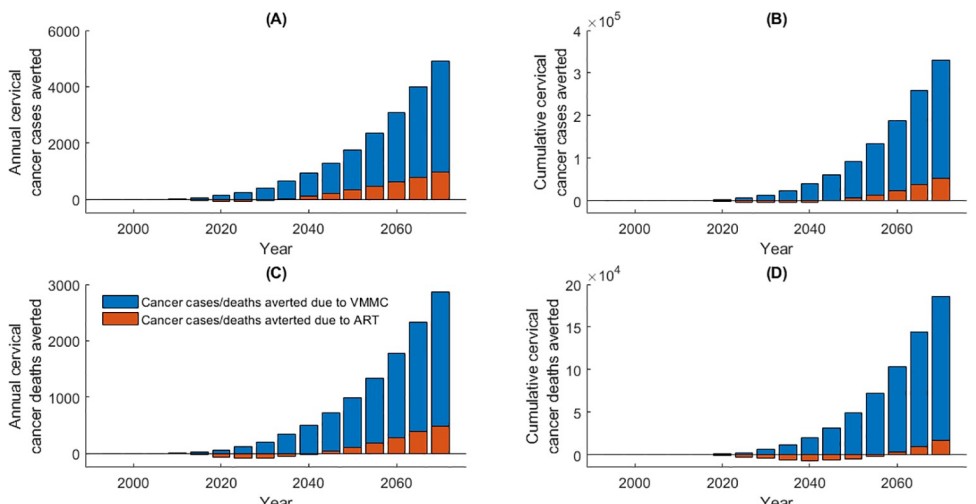

**Fig 8. (A)** Annual cervical cancer cases averted due to VMMC and ART (negative values under ART denote additional cases rather than cases averted); **(B)** cumulative cervical cancer cases averted due to VMMC and ART; **(C)** annual cervical cancer deaths averted due to VMMC and ART; **(D)** cumulative cervical cancer deaths averted due to VMMC and ART.

**Table 7. Simulated HPV and HIV prevalence for males and females in 2070 (and 2020) under five scenarios.** Note that intervention start-year occurs pre-2020 for simulated interventions.

| | HPV prevalence in 2070 (2020 value) | | HIV prevalence in 2070 (2020 value) | |
|---|---|---|---|---|
| | Males | Females | Males | Females |
| No interventions | 44.71% (47.43%) | 46.39% (49.42%) | 7.59% (5.22%) | 8.47% (6.07%) |
| VMMC only | 32.13% (34.43%) | 38.56% (42.45%) | 1.86% (3.19%) | 2.46% (4.89%) |
| VMMC and ART (baseline) | 31.90% (34.38%) | 38.28% (42.30%) | 0.41% (2.57%) | 0.59% (4.24%) |
| VMMC and target ART | 31.85% (34.38%) | 38.22% (42.30%) | 0.12% (2.57%) | 0.19% (4.24%) |
| VMMC, target ART and PrEP | 31.83% (34.38%) | 38.20% (42.30%) | 0.05% (2.57%) | 0.11% (4.24%) |

ART' scenario. In addition to this, predicted HPV prevalence in males (35.4% in 2017) under the 'VMMC and ART (baseline)' scenario showed relatively consistent agreement with observed HPV prevalence in South Africa in 2017 (40%) [61], and, predictions published by Tan et al [13]. Similarly, Tan et al predict that the cervical cancer incidence rates in KwaZulu Natal women will be approximately 31 cases per 100,000 among HIV negative women in 2070, and approximately 145 cases per 100,000 among HIV positive women (S1 File) [13]. For Tanzania, our analysis predicts 35.17 cases per 100,000 women among HIV negative women, and 220.23 cases per 100,000 among HIV positive women under the 'VMMC and ART (baseline)' scenario in 2070. In both analyses, the cervical cancer incidence rate in HIV positive women is close to five times higher than the rate among HIV negative women.

The dynamic and highly detailed nature of the model of HIV and HPV co-infection is a strength of this study. The model is stratified by sex, age, sexual activity level (including women engaging in transactional and commercial sex), HIV positivity (including disease stage and treatment status) and HPV positivity (including disease stage/detection status) for multiple

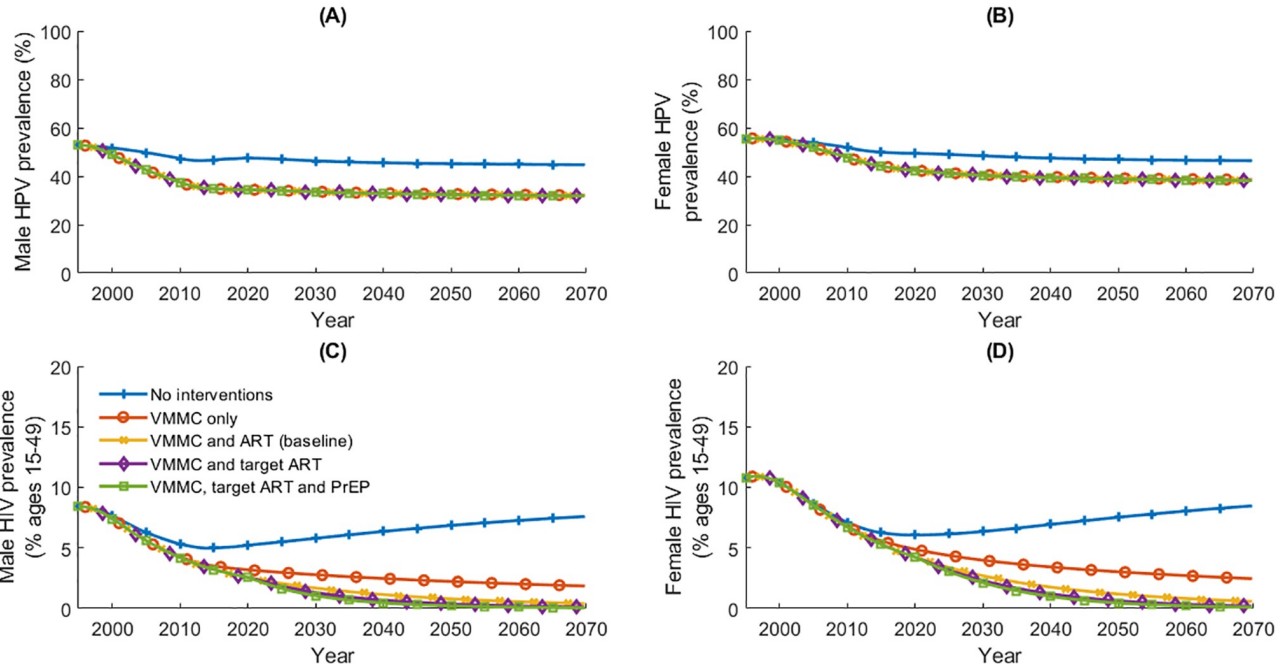

**Fig 9. (A) and (B) Male and female HPV prevalence; (C) and (D) male and female HIV prevalence from 1995 to 2070 under five intervention scenarios.**

**Table 8. Simulated age-standardised cervical cancer incidence and mortality rates per 100,000 women, for all women (and stratified by HIV positivity) in 2070 (and 2020) under five scenarios.** Rates are standardised to the WFP2015 population. [35]

| | Age-standardised cervical cancer incidence rate per 100,000 women in 2070 (2020 value) | | | age-standardised cervical cancer mortality rate per 100,000 women in 2070 (2020 value) | | |
|---|---|---|---|---|---|---|
| | **All women** | **HIV negative women** | **HIV positive women** | **All women** | **HIV negative women** | **HIV positive women** |
| **No interventions** | 55.11 (64.39) | 40.52 (44.69) | 269.89 (291.46) | 37.31 (41.99) | 28.43 (29.70) | 182.27 (198.01) |
| **VMMC only** | 39.93 (63.00) | 35.04 (42.99) | 256.25 (286.92) | 27.72 (41.47) | 24.61 (28.91) | 176.04 (197.85) |
| **VMMC and ART (baseline)** | 37.31 (65.42) | 35.17 (42.99) | 220.23 (288.57) | 26.44 (43.27) | 24.68 (28.88) | 168.01 (204.14) |
| **VMMC and target ART** | 36.28 (65.42) | 35.19 (42.99) | 208.82 (288.57) | 25.72 (43.27) | 24.70 (28.88) | 166.81 (204.14) |
| **VMMC, target ART and PrEP** | 35.82 (65.42) | 35.20 (42.99) | 192.90 (288.57) | 25.35 (43.27) | 24.71 (28.88) | 158.74 (204.14) |

HPV types; this allows exploration of disease transmission and progression dynamics in detail, accounting for herd protective effects and the protective effects of ART.

This analysis was limited by the inherent uncertainty surrounding input parameter assumptions, in particular, sexual behaviour assumptions including condom usage. In many settings which have implemented such HIV controls, a reduction in safe sex practices and an increase in other sexually transmitted infections is often observed [49–54]. A recent study into the sexual behaviour of PrEP users in Amsterdam found that daily PrEP use among HIV negative men who have sex with men (MSM) was associated with a 2–9% increase in condomless sex acts [50]; whereas another study has reported a 21% increase in risky sexual practice, and an increase in HIV incidence, among the San Franciscan MSM population since the advent of ART [53]. Sensitivity analysis findings indicate that HPV and HIV prevalence, and cervical cancer incidence and mortality are highly sensitive to variations in condom usage, therefore if condom usage trends over time vary, model predictions could substantially under-estimate or over-estimate disease burden. A number of simplifying assumptions were also made regarding the effect of ART on HPV natural history in HIV positive women; namely, we assume that

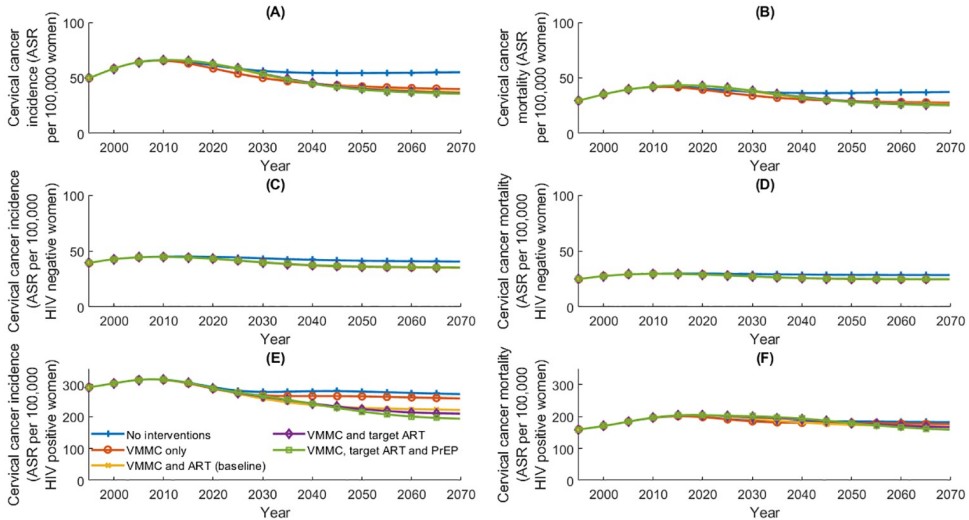

**Fig 10. (A) and (B) Age-standardised cervical cancer incidence and mortality rates among all women aged 0–99 years; (C) and (D) cervical cancer incidence and mortality rates among HIV negative women aged 0–99 years; (E) and (F) cervical cancer incidence and mortality rates among HIV positive women aged 0–99 years.** Age-standardised rates are calculated using the 2015 World Female Population [35].

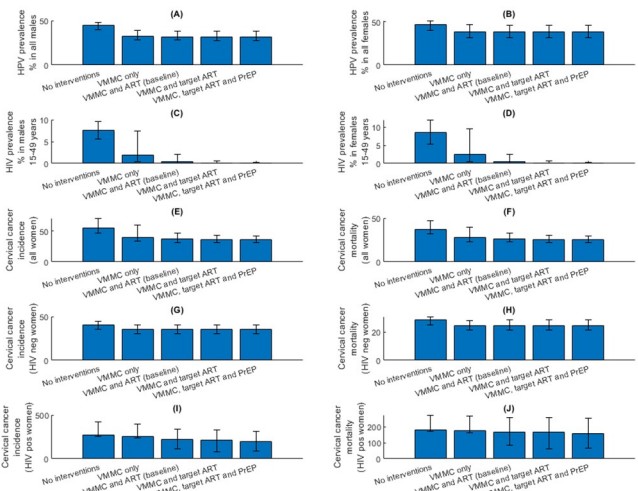

**Fig 11. (A) and (B)** Male and female HPV prevalence; **(C) and (D)** male and female HIV prevalence; **(E) and (F)** cervical cancer incidence and mortality among all women; **(G) and (H)** cervical cancer incidence and mortality among HIV negative women; **(I) and (J)** cervical cancer incidence and mortality among HIV positive women, simulated in the year 2070 (error bars correspond to the total variation generated by the sensitivity analysis).

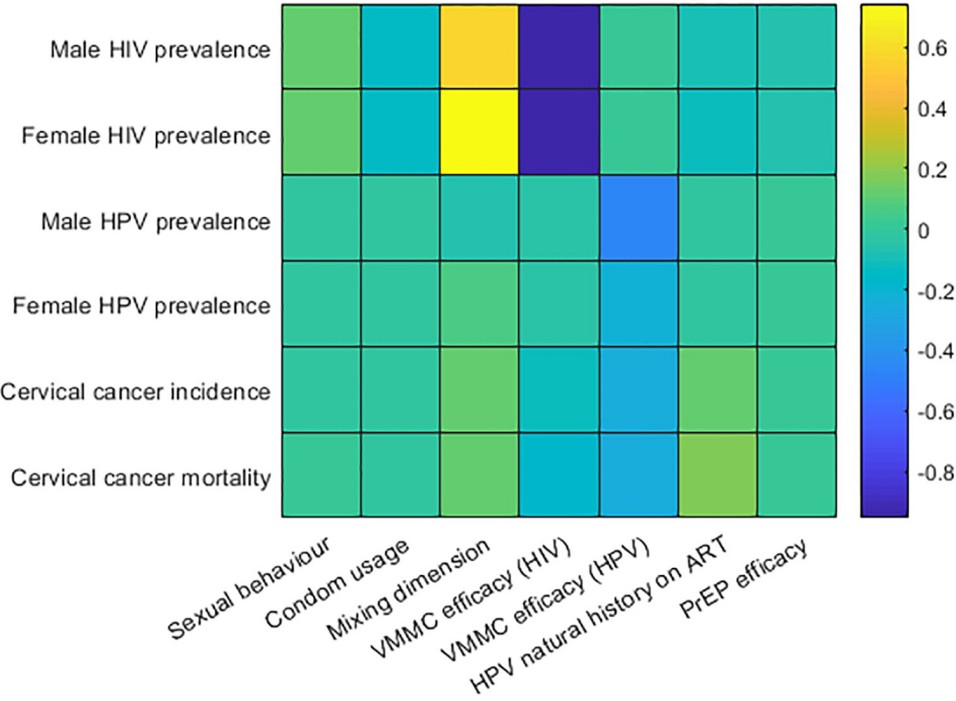

**Fig 12. Correlation strength of selected outputs (HIV and HPV prevalence for males and females, and cervical cancer incidence and mortality) against intervention and behavioural parameters varied in multivariate sensitivity analysis.** Partial rank correlation analysis was performed on the 'VMMC, target ART and PrEP' scenario, as it is the only scenario considering all modelled interventions.

ART affects all HPV types to the same degree, and, that commencing ART has the same effect on HPV natural history regardless of CD4+ count or HIV disease stage. Furthermore, the HIV transmission component of this model accounts for heterosexual transmission only, which is based on the assumption that the impact of HIV transmitted via sexual contact between men, and injection drug use, will have negligible impacts on the cervical cancer in women. Cervical cancer is an AIDS defining disease; therefore, there is a degree of uncertainty surrounding the true cervical cancer mortality rates in Tanzania, that is, whether a cervical cancer death in a HIV positive woman was attributed to cervical cancer or HIV [62]. The grouping of many individual HPV genotypes in the model, for example HPV 16/19, HPV 31/33/45/52/58 (HPV H5) and a category for "other high-risk HPV types", may impact the overall simulated HPV prevalence in addition to the overall transmission dynamics of the model; for example, in this model it is impossible to discern whether individuals are infected with only one or any combination of the HPV genotypes in each simulated HPV subgroup. This may result in an overestimation of effectiveness of interventions targeted at HPV reduction. Finally, the findings of this study must be interpreted in the context of the lengthening life expectancy in Tanzania, which is explicitly accounted for in this analysis. Due to reductions in HIV mortality in addition to other cause mortality (e.g. driven by improvements in sanitation and health care), the life expectancy at birth is expected to rise to 75 years in 2065–2070 (compared to 54 years in 1995–2000) [34]. This will necessarily result in an increased opportunity for the development of cervical cancer (and other diseases), irrespective of additional effects due to HIV treatment.

In 2019, a draft global strategy for the elimination of cervical cancer as a public health problem was released by the World Health Organisation [63]. This strategy, to be considered by the World Health Assembly in May 2020, defines that cervical cancer is eliminated as a public health problem when all countries achieve an incidence rate of less than four cases per 100 000 women per year. To achieve this target, the WHO recommends that each country implement HPV vaccination programmes whereby 90% of girls are vaccinated by the age of 15, organised cervical screening programmes whereby 70% of women are screened at least twice per lifetime, and effective management of 90% of women diagnosed with cervical pre-cancer or invasive cervical cancer [63]. While VMMC and ART can reduce the burden of cervical cancer in Tanzania in the long term, they are not sufficient to bring cervical cancer incidence beneath the threshold proposed by the WHO for cervical cancer elimination. Our finding that even under the best-case scenario the rate of cervical cancer incidence in all Tanzanian women is not reduced below 35 cases per 100,000 women per year (more than eight fold higher than the elimination threshold) demonstrates the importance and urgency of scaling up cervical cancer prevention programs, such as HPV vaccination and cervical screening, as well as HIV control, in order to avoid the situation that lives saved from HIV-related death are instead lost to cervical cancer. The WHO call for global action to eliminate cervical cancer as a public health problem is an important opportunity to galvanise and unite efforts to prevent cervical cancer in Tanzania and globally [64].

## Supporting information

**S1 File.**
(PDF)

## Acknowledgments

This research includes computations using the computational cluster Katana supported by Research Technology Services at UNSW Sydney. We also acknowledge the National Cancer

Institute–funded Cancer Intervention and Surveillance Modeling Network (CISNET) cervical cancer working group for intellectual support and feedback throughout the project.

## Author Contributions

**Conceptualization:** Michaela T. Hall, Karen Canfell, John M. Murray.

**Data curation:** Michaela T. Hall, Ruanne V. Barnabas.

**Formal analysis:** Michaela T. Hall.

**Funding acquisition:** Michaela T. Hall, John M. Murray.

**Investigation:** Michaela T. Hall, Megan A. Smith, Kate T. Simms, John M. Murray.

**Methodology:** Michaela T. Hall, Megan A. Smith, Kate T. Simms, Ruanne V. Barnabas, John M. Murray.

**Software:** Michaela T. Hall.

**Supervision:** Karen Canfell, John M. Murray.

**Validation:** Michaela T. Hall.

**Visualization:** Michaela T. Hall.

**Writing – original draft:** Michaela T. Hall.

**Writing – review & editing:** Michaela T. Hall, Megan A. Smith, Kate T. Simms, Ruanne V. Barnabas, Karen Canfell, John M. Murray.

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
