## [Decision Letter · Decision Letter 0]

15 Nov 2019

PONE-D-19-24871

The past, present and future impact of HIV prevention and control on HPV and cervical disease in Tanzania: a modelling study.

PLOS ONE

Dear Ms Hall,

first I deeply apologize but I had taken a decision when on the last week of october I had received the last report.

Evidently either I made an error or something went wrong on the PLoS ONE website.

Thank you for submitting your manuscript to PLOS ONE. After careful consideration, we feel that it has merit but does not fully meet PLOS ONE’s publication criteria as it currently stands. Therefore, we invite you to submit a revised version (*** major revisions needed ***) of the manuscript that addresses the points raised during the review process.

Based on three expert reviews I suggest major revisions.

Indeed, two of the three referees suggest directly that your manuscript needs major revisions. The third one suggests minor revisions, yet she/he list a number of  important changes.

The two referees suggesting major revisions (and also the third referee) raised a number of critical points that must carefully be dealt with.

I add some methodological and practical new comments:

A)Abstract: in the abstract you wrote "A dynamic model of HIV and HPV infection and natural history was used to simulate ". This sentence is highly uninformative since there are a huge number of possible "dynamic model " based approaches. Please, in the revised ms you mus clearly specify which class of models you used. You know, PLoS ONE is also read by people who are not scared by mathematics and statistics...

B) on line 75 you wrote that you used a "deterministic Markov model"  without mentioning any reference on this class of models. Unless you referred to "piecewise deterministic Markov models" (i.e. you made a misprint), the class of "purely" deterministic Markov models is very rare (and at the best of my knowledge of non unique definition...) and needs much more information for the readers of your manuscript. I am quite expert in many kinds of deterministic and stochastic modeling, and not only in the field of infectious diseases, but I very seldom read papers on this topic. Thus please be very detailed and include references on these purely "deterministic Markov models". References that must be both in the filed of mathematics and physics and in the field of applications concerning purely "deterministic Markov models" . For me they means "time discrete dynamical systems with some initial conditions and possibily parameters that are random variables", but maybe my definition does not fully coincide with yours.

C) Linked to both point (B) and point (F) I feel that you model must be better described (even in the main text) from the mathematical and physical viiewpoint, Be very clear in specifying what is deterministic and what is stochastic

D)Please rewrite the section (1st suppl materials) "Sexual behavior and force of infection" which in my opinion is not understandable and include it in the main text

E) The above--mentioned section "Sexual behavior and force of infection" shows that your model included important behavioral changes. It is a pity that the bibliography of your work does not include works in behavioural epidemiology of infectious diseases.

F) materials to be included in the supplementary materials: I fully agree with the observation made by one of the referees: there is too much methodological material in the supplementary materials.

40 page of supplements are pathological for a manuscript of 42 pages. Moreover, many very important points are in the two supplements: they need to be transferred in the main body text.  

PLoS ONE is a purely online journal, so without space limitations. Thus I warmly invite the authors to add as much methodological material as possible in the full body of the manuscript. 

Moreover, if you feel that some material has to stay in supplemental materials, please provide a single supplementary materials file: it is more practical for reader to download a unique file than two or more.

We would appreciate receiving your revised manuscript by Dec 20 2019 11:59PM. To enhance the reproducibility of your results, we recommend that if applicable you deposit your laboratory protocols in protocols.io, where a protocol can be assigned its own identifier (DOI) such that it can be cited independently in the future. For instructions see: http://journals.plos.org/plosone/s/submission-guidelines#loc-laboratory-protocols

We look forward to receiving your revised manuscript.

Kind regards,

Alberto d'Onofrio, Ph.D.

Academic Editor

PLOS ONE

Journal Requirements:

2. Can you please confirm whether all data and parameters used have been made available in the Supplementary materials or otherwise?

3. Please note that PLOS ONE has specific guidelines on software sharing (http://journals.plos.org/plosone/s/materials-and-software-sharing#loc-sharing-software) for manuscripts whose main purpose is the description of a new software or software package. In this case, new software must conform to the Open Source Definition (https://opensource.org/docs/osd) and be deposited in an open software archive. Please see http://journals.plos.org/plosone/s/materials-and-software-sharing#loc-depositing-software for more information on depositing your software.

In this instance, we would request information on whether the code for this manuscript is available to others, and if so, where.

I have read the journal's policy and the authors of this manuscript have the following competing interests: Karen Canfell (KC) receives salary support from the National Health and Medical Research Council (NHMRC) Australia (CDFAPP1082989). KC is a co-PI of an investigator-initiated trial of primary HPV screening in Australia (‘Compass’) which is conducted and funded by the VCS Foundation, a government-funded health promotion charity, which has received a funding contribution from Roche Molecular Systems and Roche Tissue Diagnostics, AZ, USA.

Additional Editor Comments:

Dear Authors,

two expert referees suggested major revisions and a third referee suggested minor but extensive changes.

As a consequence of the report and of my personal assessment of your manuscript, your manuscript needs a "major revision".

Please implement and reply all suggestions and questions of the three referees.

Regards,

Alberto d'Onofrio

Reviewers' comments:

Reviewer's Responses to Questions

**Comments to the Author**

1. Is the manuscript technically sound, and do the data support the conclusions?

Reviewer #1: Yes

Reviewer #2: Yes

Reviewer #3: Partly

2. Has the statistical analysis been performed appropriately and rigorously? 

Reviewer #1: Yes

Reviewer #2: Yes

Reviewer #3: N/A

3. Have the authors made all data underlying the findings in their manuscript fully available?

Reviewer #1: No

Reviewer #2: Yes

Reviewer #3: Yes

4. Is the manuscript presented in an intelligible fashion and written in standard English?

Reviewer #1: Yes

Reviewer #2: Yes

Reviewer #3: Yes

5. Review Comments to the Author

Reviewer #1: The authors present a modeling study on the effect of HIV treatment and control interventions on HPV morbidity and mortality in Tanzania over the period 1995-2070. They estimate relative reductions in HPV-related cervical cancers, cancer deaths, and overall (HIV+HPV-related) deaths, avoided via the rapid adoption of medical male circumcision observed in the country (from 8% in 1995 to 80% in 2015) and by the current and future scale-up of antiretroviral therapies and PrEP. Interstingly, they found that ART therapies may temporally increase the incidence of cervical cancer and associated mortality by increasing the life expectancy of HIV+ women who are therefore exposed for a much longer time to the risk of acquiring HPV-related cervical cancer.

The paper is well thought out, the model is carefully parameterized and thoroughly validated, and the sensitivity analysis seems sufficient to grant the robustness of results. I have only a few minor comments:

* Figure 1 needs some revision:

- the zero lines for the left and right y-axes in Fig1A are not aligned;

- the y range in Fig1A touches the axis, it would be better to leave some blank space below bars to avoid the doubt that the plot is clipped;

- I think that colors are inverted for VMMC and ART in Fig1A and 1B;

- the use of a double axis is not really necessary: I would recommend splitting each panel in two separate panels;

* showing percent reductions computed on percentages can be misleading; for example, when at line 141 the authors mention a reduction of 27.58% and 14.35% on HPV prevalence in men and women, they mean that the HPV prevalences reduces from 50% to ~36% and from 55% to ~47%; but it could be misinterpreted as a reduction from 50% to 50-27.58 = 22.62% and 55-14.35= 40.65%. To avoid possible misunderstandings, I recommend to declare final prevalences, (possibly indicating relative reductions with expressions such as "by approximately one fourth / one seventh").

* age-group limits vary by type of estimate: 20-64 years old for HPV prevalence, 15-49 for HIV prevalence, 0-80 for cervical cancer incidence. I believe this was done to comply with parametrization/calibration data: if so, can the authors state it explicitly? In addition, it could be useful to provide age-specific prevalence/incidence predicted by the model at different time points to give a better idea on which age groups are most impacted by interventions;

* the reduction on condom usage observed between 2011 (58%) and 2016 (37%) is remarkable and worrying. Considering that the trend had been consistently increasing in the previous years, can the author comment on this sudden drop? Could it be related to a measurement error/spurious data? It would be interesting to explore a scenario where the condom usage levels are reverted to at least the 2011 levels, e.g. through awareness campaigns;

* ASR (age-standardized rate?) is undefined in y labels of Fig4-6; in Fig the y lab of Fig6, a closed parenthesis is missing;

* the description of results could be improved by reducing the text commenting figures and collecting the reported values in a table comparing scenario results for different target variables. For example, for Fig. 2 and 5, I would not spend too much text in commenting a substantially equivalent prediction for all intervention scenarios (differences in scenario results are likely to be non-significant with respect to uncertainties in parameter values). Similar simplifications can be done for other Figures, allowing the reader to focus his/her attention on really different dynamics and mechanisms underlying them;

* at line 335, the authors mention an elimination threshold proposed by the WHO for cervical cancer: how much is it? Considering that HPV vaccination is already implemented in the code, why don't the authors use it to provide scenarios at different coverage levels, in such a way to suggest possible targets for a vaccination campaign aimed at elimination?

* I expect the increase in life expectancy projected for Africa in the next decades to have a very dramatic effect on chronic infections such as HPV and HIV, contributing with a similar mechanism as ART (the reduction of competing mortality) to the rise of cervical cancers and possibly HIV prevalence. If the authors can't include projected mortality rates in the model, they should discuss this as a model limitation and qualitatively speculate on the likely impact of this issue.

Reviewer #2: The paper is an interesting one because it sets-up a relevant issue namely, which are the implications that successful control measures against HIV in Sub- Saharan Africa (SSA) might have for other ST infections representing an important public health burden, first of all HPV.

However, the manuscript should be improved in a number of directions, departing from the introduction.

First, the description of the context of HIV-HPV in Tanzania is somewhat scanty. For example, it is mentioned at L63 that incidence of CC in Tanzania was among the highest in the world, are there explanations for such a high incidence rate of CC in Tanzania? This cannot simply be HIV- and underlying causes, which is the leit-motiv of the manuscript, given that the Tanzanian HIV epidemic is very far from being among the worst HIV epidemic in SSA.

Also, some lines framing the state of HIV & HPV in Tanzania within the broader framework of such diseases in Sub-Saharan Africa as a whole would be very welcome by potential readers.

Parallel to this also the state and perspective of current interventions against HIV in Tanzania should be introduced. I am surprised authors do not consider condom use at all, which is still dramatically suboptimal in SSA. Similarly, nothing is said about interventions against HPV, particularly about the access to vaccination (unless this is in the Sup Mat).

L32 (and related sentences in the Results) The remark on the enhancing effect that ART will have on CC is not surprising from a demographic viewpoint: in the history of mankind any control reducing the impact of a deadly infective cause will unavoidably open room for other mortality causes.

L68. (and related sentences in the Results) This sentence is somewhat ambiguous: it is not clear whether the paper also aims to represent a model of the impact of HIV intervention on HIV itself. According to what stated here the answer seems to be "no", that is the role of modelling HIV seems to be purely “of service” for modelling its output on HPV. Is this correct? I understand that the model of HIV is quite oversimplified which does not make it a good model for predicting the HIV epidemics per se, but in this case one should perhaps motivate why then to model HIV at all, rather than, simply taking the most parsimonious choice namely, postulating some direct effects that HIV control measures in place in Tanzania might have for HPV circulation and eventually for progression to CC. This point should be considered carefully.

Model presentation is very rapid, with all details relegated to the Sup Mat. However, readers would like to have the possibility to judge about the goodness of the adopted model (it is by this goodness that we can judge the relevance of the model predictions) without the need to continuously switch to the Sup Mat. Therefore, I recommend that the part of the results that deal with the estimation of critical model parameters (basically, the “Calibration results” of Sup Mat 1) are reported in the main text in the Results section. Yet, the Sup Mat 1 should be better organised. There are no model equations, the demographic part is a bit unclear (seemingly the authors use a variable population, but this is totally contrasting with the idea that the population is “uniformly distributed over 5-years age groups” etc), information on mixing patterns and related references should be made clear and usable by readers.

Specific points

L17 “59.1 per 100,000” per year

L18 "the impact that intervention aimed at controlling HIV"

L21 It is not made clear to readers that these are the three main interventions against HIV currently in place in Tanzania.

L26 It is unclear to what the figure of 24,967 total deaths actually refers to. From the results it appears that it is the sum of prevented deaths by AIDS and CC. This should be clarified.

L54 "reduce HIV-1 acquisition"? presume you mean "hazard of acquisition" but still is unclear, which unit are you referring to ? per single sexual act ? or what else ?

L56 prevelance

L64: (19) and (20) are independent studies so cit 19 should be reported after "with HIV"

Description “dual HIV-HPV” is often used. I suggest to use some cleared description.

L98 “For each scenario, we estimated HPV and HIV prevalence”. At least over 2020-2070 you should say “we projected” (based on the model hypotheses)

I suggest to put Table 1 in concise form avoiding long verbal descriptions (that can be put in the caption) in table entries.

Reviewer #3: The authors of this paper developed a deterministic transmission-dynamic model of HPV and HIV transmission, and natural history of HPV-related cervical cancer and HIV, to estimate the impact of actual and scaled-up HIV control actions on past and future trends of cervical cancer incidence and mortality in Tanzania. HIV control actions include voluntary medical male circumcision (VMMC), anti-retroviral therapy (ART), and pre-exposure prophylaxis. The model parameters were fixed based on the literature or calibrated to empirical data. Five scenarios of HIV control actions (no control, only VMMC, actual control with VMMC and ART, and three scaled-up control scenarios) were elaborated and compared in terms of cervical cancer incidence and mortality between 1995 and 2070. Age-standardized incidence of cervical cancer and mortality due to cervical cancer, number of deaths due to cervical cancers, number of cases of cervical cancer, and total number of death averted, HPV prevalence, HIV prevalence, were used as outcomes for comparison.

This study shows that interventions aimed at HIV prevention can have major indirect effect on cervical cancer. This important effect would be missed in a typical effectiveness analysis which does not include HPV infection and transmission. The authors’ usage of a model of HPV/HIV transmission and development of subsequent cervical cancer is necessary if HIV infection can affect both HPV transmission and HPV natural history of disease. The mathematical model used appears adequate overall with a potential limitation regarding the grouping of HPV-genotypes (e.g., a single group for all cross-protective types). However, the calibration does not account sufficiently for the substantial parameter uncertainty related to the complicated and still uncertain interaction between HIV and HPV, and such uncertainty is not visible in the results presented. Hence, the present study requires some methodological adjustment but is a potentially important contribution to the assessment of HIV prevention effectiveness in Tanzania and other sub-Saharan countries of similar context.

Major comments

The estimation of the effect of parameters uncertainty on the results is unclear or inadequate in this study. To my understanding, two sensitivity analyses were done. In the first analysis, a subset of the parameters is varied and their impact on calibration targets is assessed: this cannot be used to estimate parameters uncertainty on the study outcomes (HIV control actions effectiveness). Even if varying a parameter does not have an impact on calibration target this does not mean that the parameter does not have an impact on effectiveness. For example, different combinations of parameter can lead to the same fit of pre intervention incidence, but produce sharp differences in post intervention effectiveness. In the second analysis, parameters related to the effect of HIV control actions (e.g., effect of circumcision on transmission) are varied and their impact on the study outcomes are ascertained, but this analysis does not include other uncertain parameters such as those related to sexual behavior. In the presentation of results, a single number is given with no uncertainty interval. It is highly unlikely that the numbers given are precise enough to justify omitting an uncertainty interval.

The methodology of this modelling study is complex and is difficult to explain thoroughly in a short article. However, some essential information is missing and was moved to the supplementary materials: 1) information on calibration methods, 2) calibration data related to sexual behavior. Overall, although a lot of information on the methods are in Appendices, there should still be sufficient information in the core paper for readers to understand the approach and key simplifying assumptions.

To help understand the results, they should include the overall impact of HIV interventions partitioned as the effect due to HIV reduction only plus the secondary effect on cervical cancer. This should be done with absolute and relative measures (number of deaths and mortality rates). Presenting only the absolute number of deaths averted is insufficient. Absolute numbers can be diluted by other effects such as population growth. Presenting both relative and absolute effects allow both to understand the underlying results (relative impact) and provide numbers for policy makers (absolute). In addition, relative numbers may help with generalizability (or discussion of generalizability) of the results to other settings.

The limits related to grouping HPV-genotypes should be at the minimum mentioned. Grouping HPV-types is known to produce “super-bug” in the general population. In the context of this study with an important HIV-infected population, it is not immediately clear what would be the impact of grouping genotypes. Given that people living with HIV have a high prevalence of HPV co-infections, grouping types may have an impact on results as individuals in the model cannot be co-infected by more than 3 super types (what happens with competing risks towards HPV-related disease, transmission dynamics, etc?).

I think more explanation of the following conclusions is needed: “These results demonstrate that HIV control measures will have a substantial effect in reducing HIV-related death in Tanzania, but will have an unintended consequence of increasing cervical cancer incidence and mortality in HIV-positive women for the next few decades, as a result of their increased life expectancy.” “These findings demonstrate the importance and urgency of scaling up cervical cancer prevention programs, such as HPV vaccination and cervical screening, as well as HIV control, in order to avoid the situation that lives saved from HIV-related death are instead lost to cervical cancer.”

In the results, the age-standardized mortality due to cervical cancer decreases after HIV prevention. Hence, it could be argued that HIV prevention has a positive effect on mortality rates due to cervical cancer. The increase is in absolute number of deaths due to cervical cancer, because deaths have been prevented and people are living longer. It would be good to have more discussion on this, so that readers better understand the implications. If childhood mortality is prevented, there will be more deaths among adults. If deaths are prevented among young adults, more deaths will occur among older adults, etc. Lives saved from HIV will results in substantial life years gained, which should not be mistakenly interpreted as something negative. Rather that it will have consequences on the number of cervical cancer cases, and thus provide even more justification for cervical cancer prevention. But not only for cervical cancer prevention as it will increase the number of deaths due to other prevalent diseases among people living with HIV.

 

Minor comments

In the No intervention scenario, condom usage declines between 2011 and 2016. However, could this be partially due to the HIV prevention measures? The authors mention that HIV prevention can affect sexual behavior and condom usage. If so, shouldn’t the decline in condom usage be ignored in the No intervention scenario?

In First Supplementary Appendix to the article:

P. 3, last paragraph: "The number of high-risk sexual interactions is defined as the total number of sexual interactions which could potentially result in HIV transmission." This sentence needs to be re-written, unless the assumption is that low-risk sexual interaction can’t result in HIV transmission.

6. PLOS authors have the option to publish the peer review history of their article (what does this mean?). If published, this will include your full peer review and any attached files.

Reviewer #1: Yes: Giorgio Guzzetta

Reviewer #2: No

Reviewer #3: No

---

## [Author Response · Author response to Decision Letter 0]

15 Jan 2020

Note that I have uploaded a "Response to Reviewers" document that is easier to follow. 

Dear Dr d'Onofrio,

Thank you for considering our manuscript titled The past, present and future impact of HIV prevention and control on HPV and cervical disease in Tanzania: a modelling study. We would like to thank the reviewers and yourself for the comments, which we have endeavoured to incorporate. 

Please note that the methods section of this manuscript has been almost entirely re-written to improve on the level of detail and clarity, as requested by yourself and the reviewers. The results section has also been substantially changed to include additional information about model calibration/validation outcomes. While we have endeavoured to eliminate/reduce the size of the supplementary appendix, it is still necessary to have one due to the large number of equations and sizable data tables. 

It has been necessary to re-run the model with altered input assumptions in response to review comments. Namely, we have re-run the model to incorporate the projected future life-expectancy in Tanzania (correcting for improvements due to lowered HIV and cervical cancer mortality) and we have combined the two separate sensitivity analyses into one larger sensitivity analysis. Furthermore, we have edited the base population structure used for the age-standardisation in the presentation of cervical cancer incidence and mortality rates; this was done for consistency with recent literature published in the field of HPV/cervical cancer modelling (Simms et al 2019 https://doi.org/10.1016/S1470-2045(18)30836-2). 

Please see our response to all comments in the below. 

Best regards,

Michaela Hall on behalf of all co-authors

Comments from the editor

A) Abstract: in the abstract you wrote "A dynamic model of HIV and HPV infection and natural history was used to simulate ". This sentence is highly uninformative since there are a huge number of possible "dynamic model " based approaches. Please, in the revised ms you must clearly specify which class of models you used. You know, PLoS ONE is also read by people who are not scared by mathematics and statistics...

For clarity, the methods section of the abstract (p2 line 22) has been edited to read “A deterministic transmission-dynamic compartment model”. 

B) on line 75 you wrote that you used a "deterministic Markov model” without mentioning any reference on this class of models. Unless you referred to "piecewise deterministic Markov models" (i.e. you made a misprint), the class of "purely" deterministic Markov models is very rare (and at the best of my knowledge of non-unique definition...) and needs much more information for the readers of your manuscript. I am quite expert in many kinds of deterministic and stochastic modelling, and not only in the field of infectious diseases, but I very seldom read papers on this topic. Thus, please be very detailed and include references on these purely "deterministic Markov models". References that must be both in the field of mathematics and physics and in the field of applications concerning purely "deterministic Markov models”. For me they mean "time discrete dynamical systems with some initial conditions and possibly parameters that are random variables", but maybe my definition does not fully coincide with yours.

Thank you for pointing this out. The line now reads (for clarity and consistency with the abstract) “A deterministic transmission-dynamic compartment model”. This can be found on line 90 (page 5). 

C) Linked to both point (B) and point (F) I feel that you model must be better described (even in the main text) from the mathematical and physical viewpoint, be very clear in specifying what is deterministic and what is stochastic

Please note that the methods section of the model has been substantially re-written to address editor and reviewer concerns over the level of detail provided. In order to more thoroughly (and clearly) describe the model, a new Table 1 has been inserted (p7 line 111). This table specifies that model compartments can be described as the cartesian product over nine vectors, labelled in the table as (A) through to (I) every compartment, and notes that all state transitions are deterministic. Note that the description for each process simulated by the model (e.g. births, deaths, ageing, infection, etc.) has been moved from the supplementary appendix to the main text. 

D) Please rewrite the section (1st suppl materials) "Sexual behaviour and force of infection" which in my opinion is not understandable and include it in the main text

As requested, this section has been entirely re-written for clarity, and relocated to the main text (see lines 146 to 195 over pages 10 to 12). The force of infection is now described using equations, for a mathematically inclined readership (see response to reviewer 2). Note that input parameter assumptions regarding calculation of the force of infection remain in the supplementary appendix due to it being a very large table spanning multiple pages. 

E) The above--mentioned section "Sexual behavior and force of infection" shows that your model included important behavioral changes. It is a pity that the bibliography of your work does not include works in behavioural epidemiology of infectious diseases.

 Available evidence suggests that the advent of effective HIV treatments (ART) and preventions (PrEP) have facilitated the observed (and modelled) behavioural disinhibition. This evidence has been discussed and referenced in the discussion on line 431, page 31. 

“This analysis was limited by the inherent uncertainty surrounding input parameter assumptions, in particular, sexual behaviour assumptions including condom usage. In many settings which have implemented such HIV controls, a reduction in safe sex practices and an increase in other sexually transmitted infections is often observed(49-54). A recent study into the sexual behaviour of PrEP users in Amsterdam found that daily PrEP use among HIV negative men who have sex with men (MSM) was associated with a 2-9% increase in condomless sex acts(50); whereas another study has reported a 21% increase in risky sexual practice, and an increase in HIV incidence, among the San Franciscan MSM population since the advent of ART(53). Sensitivity analysis findings indicate that HPV and HIV prevalence, and cervical cancer incidence and mortality are highly sensitive to variations in condom usage, therefore if condom usage trends over time vary, model predictions could substantially under-estimate or over-estimate disease burden.”

F) materials to be included in the supplementary materials: I fully agree with the observation made by one of the referees: there is too much methodological material in the supplementary materials. 40 page of supplements are pathological for a manuscript of 42 pages. Moreover, many very important points are in the two supplements: they need to be transferred in the main body text. PLoS ONE is a purely online journal, so without space limitations. Thus I warmly invite the authors to add as much methodological material as possible in the full body of the manuscript. Moreover, if you feel that some material has to stay in supplemental materials, please provide a single supplementary materials file: it is more practical for reader to download a unique file than two or more.

We thank the editor and reviewers for this feedback and have taken this opportunity to move substantial portions of the supplementary appendices to the main text (see responses above). In addition, the supplementary material has been combined into a single file. Note that there still exists a supplementary appendix, which is now reserved for very large tables equations only. 

Reviewer #1: The authors present a modeling study on the effect of HIV treatment and control interventions on HPV morbidity and mortality in Tanzania over the period 1995-2070. They estimate relative reductions in HPV-related cervical cancers, cancer deaths, and overall (HIV+HPV-related) deaths, avoided via the rapid adoption of medical male circumcision observed in the country (from 8% in 1995 to 80% in 2015) and by the current and future scale-up of antiretroviral therapies and PrEP. Interestingly, they found that ART therapies may temporally increase the incidence of cervical cancer and associated mortality by increasing the life expectancy of HIV+ women who are therefore exposed for a much longer time to the risk of acquiring HPV-related cervical cancer.

The paper is well thought out, the model is carefully parameterized and thoroughly validated, and the sensitivity analysis seems sufficient to grant the robustness of results. I have only a few minor comments:

* Figure 1 needs some revision:

- the zero lines for the left and right y-axes in Fig1A are not aligned;

- the y range in Fig1A touches the axis, it would be better to leave some blank space below bars to avoid the doubt that the plot is clipped;

- I think that colors are inverted for VMMC and ART in Fig1A and 1B;

- the use of a double axis is not really necessary: I would recommend splitting each panel in two separate panels;

Thank you for pointing out these issues and providing advice on increasing the readability of this figure. We have made the recommended changes to figure 1, which is now called “Figure 8”. 

* showing percent reductions computed on percentages can be misleading; for example, when at line 141 the authors mention a reduction of 27.58% and 14.35% on HPV prevalence in men and women, they mean that the HPV prevalences reduces from 50% to ~36% and from 55% to ~47%; but it could be misinterpreted as a reduction from 50% to 50-27.58 = 22.62% and 55-14.35= 40.65%. To avoid possible misunderstandings, I recommend to declare final prevalences, (possibly indicating relative reductions with expressions such as "by approximately one fourth / one seventh").

Thank you for pointing this out. We have now included tables in the results section which report on the absolute values of model outcomes (including HIV/HPV prevalence’s). This was to increase readability/clarity (as per your point here) and reduce the quantity of text in the results write-up (as per your point below). In particular, the prevalence of HIV and HPV are now explicitly reported in Table 7 (page 26, line 341). 

* age-group limits vary by type of estimate: 20-64 years old for HPV prevalence, 15-49 for HIV prevalence, 0-80 for cervical cancer incidence. I believe this was done to comply with parametrization/calibration data: if so, can the authors state it explicitly? 

This is correct, the difference in age group limits for HIV prevalence, HPV prevalence and cervical cancer incidence was driven by what was reported in the available observed data (i.e. we utilised data in the age-groups reported by various sources). We have now stated this explicitly on page 9 (lines 120-122). “Note that different groupings in the age-range of calibration/validation results are due to variation in reported age-ranges in the observed data.” 

In addition, it could be useful to provide age-specific prevalence/incidence predicted by the model at different time points to give a better idea on which age groups are most impacted by interventions;

While it may be of some interest to provide age-specific breakdowns at various timepoints for all interventions, we are of the opinion that it is beyond the scope of the intended work (i.e. to estimate the impact of HIV prevention on cervical cancer over a long period of time). However, the model is calibrated to age-specific data, as presented in the results. 

* the reduction on condom usage observed between 2011 (58%) and 2016 (37%) is remarkable and worrying. Considering that the trend had been consistently increasing in the previous years, can the author comment on this sudden drop? Could it be related to a measurement error/spurious data? It would be interesting to explore a scenario where the condom usage levels are reverted to at least the 2011 levels, e.g. through awareness campaigns;

The sudden drop in condom usage over time is consistent with available evidence suggesting that the introduction of effective treatment (ART) promotes a level of behavioural disinhibition. For a more detailed response to this point, please see our reply above to the editor’s point (E). Note that variation in condom usage over time is also considered in the sensitivity analysis. 

* ASR (age-standardized rate?) is undefined in y labels of Fig4-6; in Fig the y lab of Fig6, a closed parenthesis is missing;

Thank you pointing out this omission. The age-standardised rate has now been defined in all relevant figures. In addition, the calculation for the ASR is described on page 16-17 (lines 255 to 258). “Here, the age-standardised rate is a weighted mean of the age-specific rates where weights (summing to one) are derived from the estimated world female population aged 0-99 years (38), and is presented per 100,000 women.”

* the description of results could be improved by reducing the text commenting figures and collecting the reported values in a table comparing scenario results for different target variables. For example, for Fig. 2 and 5, I would not spend too much text in commenting a substantially equivalent prediction for all intervention scenarios (differences in scenario results are likely to be non-significant with respect to uncertainties in parameter values). Similar simplifications can be done for other Figures, allowing the reader to focus his/her attention on really different dynamics and mechanisms underlying them;

Thank you for this suggestion; we have now included data tables (table 6 on page 25, table 7 on page 26 and table 8 on pages 27-28) to summarise the model outputs and substantially reduce text. 

* at line 335, the authors mention an elimination threshold proposed by the WHO for cervical cancer: how much is it? Considering that HPV vaccination is already implemented in the code, why don't the authors use it to provide scenarios at different coverage levels, in such a way to suggest possible targets for a vaccination campaign aimed at elimination?

The World Health Organisation has released a draft strategy which proposes that cervical cancer could be considered eliminated as a public health problem in a country if the age-standardised rates of cervical cancer falls below 4 cases per 100,000 women, high coverage of human papillomavirus (HPV) vaccination, cervical screening and treatment of precancer, treatment of invasive cancer, and palliative care is implemented. This has now been clarified in the discussion section of the manuscript (lines 462-470 p32-33):

“In 2019, a draft global strategy for the elimination of cervical cancer as a public health problem was released by the World Health Organisation(63). This strategy, due to be assessed by the World Health Assembly in May 2020, defines that cervical cancer is eliminated as a public health problem when all countries achieve an incidence rate of less than four cases per 100 000 women per year. To achieve this target, the WHO recommends that each country implement HPV vaccination programmes whereby 90% of girls are vaccinated by the age of 15, organised cervical screening programmes whereby 70% of women are screened at least twice per lifetime, and effective management of 90% of women diagnosed with invasive cervical cancer(63).”

Addressing HPV vaccination strategies to eliminate cervical cancer in Tanzania is outside the scope of this particular paper (which addresses the unintended impacts of HIV) and will be the focus of a separate piece of work. 

* I expect the increase in life expectancy projected for Africa in the next decades to have a very dramatic effect on chronic infections such as HPV and HIV, contributing with a similar mechanism as ART (the reduction of competing mortality) to the rise of cervical cancers and possibly HIV prevalence. If the authors can't include projected mortality rates in the model, they should discuss this as a model limitation and qualitatively speculate on the likely impact of this issue.

We are pleased to say that we have re-run the model incorporate the projected age-specific annual mortality rates for Tanzania. Therefore, our model accounts for increasing life expectancy in general (and not just due to decreasing HIV death and cervical cancer mortality). We have clarified this point in lines 139-140 (p10) “Age-and-year-specific natural mortality rates specified using the projected year-on-year life tables reported by the United Nations Population Division(31)”. 

Reviewer #2: The paper is an interesting one because it sets-up a relevant issue namely, which are the implications that successful control measures against HIV in Sub- Saharan Africa (SSA) might have for other ST infections representing an important public health burden, first of all HPV.

However, the manuscript should be improved in a number of directions, departing from the introduction.

First, the description of the context of HIV-HPV in Tanzania is somewhat scanty. For example, it is mentioned at L63 that incidence of CC in Tanzania was among the highest in the world, are there explanations for such a high incidence rate of CC in Tanzania? This cannot simply be HIV- and underlying causes, which is the leit-motiv of the manuscript, given that the Tanzanian HIV epidemic is very far from being among the worst HIV epidemic in SSA.

Thank-you. While HIV positivity does increase the likelihood that an HPV infection will progress to cervical cancer, it is not accurate to say that a high prevalence of HIV will necessarily result in high incidence of cervical cancer. Human papillomavirus infection is the causative agent of cervical cancer, but in addition to HPV exposure (and the prevalence of various HPV sub-types, which differs by geographic region),cervical cancer incidence depends on exposure to the known co-factors in HPV progression (multiparity, smoking, age at first full term pregnancy, and use of oral contraceptives, as well as HIV) and also whether or not any level of opportunistic cervical screening is done. Also note that there are major limitations in the availability of IARC certified registry data in SSA so caution should be used in comparisons between countries.

Also, some lines framing the state of HIV & HPV in Tanzania within the broader framework of such diseases in Sub-Saharan Africa as a whole would be very welcome by potential readers.

Parallel to this also the state and perspective of current interventions against HIV in Tanzania should be introduced. I am surprised authors do not consider condom use at all, which is still dramatically suboptimal in SSA. Similarly, nothing is said about interventions against HPV, particularly about the access to vaccination (unless this is in the Sup Mat).

We thank the reviewer for their suggestion. Please note that condom usage is considered and explicitly modelled in this analysis, please refer to Table s1 in the supplementary appendix. Some additional framing context on HPV and HIV in Tanzania and sub-Saharan Africa has been included in the introduction. See lines 66-83 on pages 4-5 which now read:

“The United Republic of Tanzania has a high burden of both HIV and cervical cancer. It is estimated that in 2018, 5.5% of Tanzanian women aged 15-49 years were living with HIV(22), while the incidence of cervical cancer was among the highest globally, at 59.1 cases diagnosed per 100,000 women (9,771 cervical cancers detected) in 2018(23). The 2018 incidence rates of cervical cancer in Southern Africa and Eastern Africa were 43.1 cases per 100,000 women per year, and 40.1 cases per 100,000 women per year, respectively(23). Tanzania is within the sub-Saharan African region, which in 2018 contained 53% of all people living with HIV globally and had an estimated HIV prevalence among adults aged 15-49 years of 7%(24, 25). The two main interventions against HIV currently in place in Tanzania are ART for HIV positive individuals, and VMMC, which are both being actively scaled up(26); while the Tanzanian Ministry of Health recommends PrEP use for those at significant risk of HIV acquisition, scale up of access to PrEP has been minimal(27). In light of Tanzania’s high burden of cervical cancer and the known impact of HIV, it is important to assess the impact of HIV control interventions that are currently being scaled up (ART and VMMC) or considered (PrEP) on not only HIV incidence and prevalence, but also rates of cervical cancer incidence and mortality. While there exists significant variation in national laws pertaining to sexual identity and orientation, sex-work and access to contraception across the African continent which affects local rates of sexually transmitted diseases(24), the relative impact of HIV interventions on cervical cancer incidence rates in Tanzania is likely to be broadly representative of the region.”

L32 (and related sentences in the Results) The remark on the enhancing effect that ART will have on CC is not surprising from a demographic viewpoint: in the history of mankind any control reducing the impact of a deadly infective cause will unavoidably open room for other mortality causes.

We agree but suggest that while this remark (and related remarks) may be obvious to the reviewer, we believe that it is important to include the statement anyway. 

L68. (and related sentences in the Results) This sentence is somewhat ambiguous: it is not clear whether the paper also aims to represent a model of the impact of HIV intervention on HIV itself. According to what stated here the answer seems to be "no", that is the role of modelling HIV seems to be purely “of service” for modelling its output on HPV. Is this correct? I understand that the model of HIV is quite oversimplified which does not make it a good model for predicting the HIV epidemics per se, but in this case one should perhaps motivate why then to model HIV at all, rather than, simply taking the most parsimonious choice namely, postulating some direct effects that HIV control measures in place in Tanzania might have for HPV circulation and eventually for progression to CC. This point should be considered carefully.

The primary focus of this manuscript is to quantify the impact that HIV prevention and control to date has already had and is anticipated to have in the future on cervical cancer incidence. So, the HIV modelling is largely “in service to” the cervical cancer incidence modelling. As cervical cancer (which can occur only in women) is the primary outcome, and cervical cancer is caused by HPV (transmitted only via sexual contact) the HIV components do not account for MSM or injection-drug users, as the impact of these on the primary outcome of this analysis is expected to be negligible. The level of detail included in the HIV components of this model are largely consistent with that of previously published models assessing elimination of HIV in Africa (PLoS Med. 2012;9(7):e1001245. doi: 10.1371/journal.pmed.1001245. Epub 2012 Jul 10). Therefore, we respectfully disagree with the statement that the HIV components of this model are “oversimplified” for the purposes of modelling cervical cancer. Nonetheless, this has been addressed as a limitation in the discussion (lines 445-447 on p32). 

“Furthermore, the HIV transmission component of this model accounts for heterosexual transmission only, which is based on the assumption that the impact of HIV transmitted via sexual contact between men, and injection drug use, will have negligible impacts on the cervical cancer in women.”

Model presentation is very rapid, with all details relegated to the Sup Mat. However, readers would like to have the possibility to judge about the goodness of the adopted model (it is by this goodness that we can judge the relevance of the model predictions) without the need to continuously switch to the Sup Mat. Therefore, I recommend that the part of the results that deal with the estimation of critical model parameters (basically, the “Calibration results” of Sup Mat 1) are reported in the main text in the Results section. 

Thank you for pointing this out, we have made the recommended in text changes. Please see our response to comments above from the editor and reviewer 1 on this point.

Yet, the Sup Mat 1 should be better organised. There are no model equations, the demographic part is a bit unclear (seemingly the authors use a variable population, but this is totally contrasting with the idea that the population is “uniformly distributed over 5-years age groups” etc), information on mixing patterns and related references should be made clear and usable by readers.

A large proportion of the supplementary appendix has been re-located to the main text, including the re-writing the section pertaining to sexual behaviour and force of infection (starting line 146 on page 10). To clarify simulation of population demography (specifically ageing) we have re-worded some text in the subsection titled “demography” (now also in the main text). See line 127-134 p 9

“The youngest simulated age group is 5-9 years, therefore recruitment represents the number of children born who survive to age five, and accounts for the age- and year-specific fertility rates of the simulated female population, as well as infant mortality. The probability in each timestep of any individual ageing to the next five-year age-group is calculated using the number of single-year ages the age group, and the number of model iterations per year. For example, we assume that 1/5 of individuals in the 10-14 year age-group will turn 15 in any given year, and since there are four timesteps simulated per year, the probability of ageing from the 10-14 year group to the 15-19 year group is 1/5×1/4=0.05. ”

Specific points

L17 “59.1 per 100,000” per year

Thank you for pointing this out, we have made the recommended in text changes (see line 17 p 2). 

L18 "the impact that intervention aimed at controlling HIV"

Line 18-19 p 2 now reads “… to quantify the impact that interventions aimed at controlling HIV …”

L21 It is not made clear to readers that these are the three main interventions against HIV currently in place in Tanzania.

To clarify this point, the following text has been inserted on page 5, lines 73-76. 

“The two main interventions against HIV currently in place in Tanzania are ART for HIV positive individuals, and VMMC, which are both being actively scaled up(26); while the Tanzanian Ministry of Health recommends PrEP use for those at significant risk of HIV acquisition, scale up of access to PrEP has been minimal(27).”

L26 It is unclear to what the figure of 24,967 total deaths actually refers to. From the results it appears that it is the sum of prevented deaths by AIDS and CC. This should be clarified.

Thank you for pointing this out. The figure has now been defined (in Table 6) as “deaths due to HIV or cervical cancer prevented cumulatively”. 

L54 "reduce HIV-1 acquisition"? presume you mean "hazard of acquisition" but still is unclear, which unit are you referring to ? per single sexual act ? or what else ?

Three randomised controlled intervention trials assessed the capacity of male circumcision to prevent HIV acquisition (Auvert PLoS Med 2005; Baily Lancet 2007; Gray Lancet 2007). Here, uncircumcised men were randomised to the intervention arm (VMMC) or the control arm (no VMMC). For two of these studies the follow-up time was 24 months, and for one study the follow-up time with 18 months. The average relative reduction in risk for HIV acquisition in circumcised men found through these trials was calculated to be 60%. 

To clarify the meaning of this, the text in the introduction has been altered (lines 59-61 on page 4) to read

“In particular, male circumcision has been shown to reduce the risk of HIV-1 acquisition over a time-period of 18-24 months in heterosexual men by at least 60%, and, reduce HPV prevalence among heterosexual men by 63%(15-19).”

L56 prevelance

“Prevalence” is now spelled correctly.

L64: (19) and (20) are independent studies so cit 19 should be reported after "with HIV"

Description “dual HIV-HPV” is often used. I suggest to use some cleared description.

The citations have now been separated (see lines 66-69 on pages 4-5).

L98 “For each scenario, we estimated HPV and HIV prevalence”. At least over 2020-2070 you should say “we projected” (based on the model hypotheses)

Please see our changes to line 452-454 on page 16

“For each scenario, we estimate HPV and HIV prevalence, and cervical cancer incidence and cervical cancer mortality (stratified by HIV positivity) from 1995-2020, and project these outcomes based on model hypotheses from 2021 to 2070”. 

I suggest to put Table 1 in concise form avoiding long verbal descriptions (that can be put in the caption) in table entries.

We have significantly reduced the amount of text contained in this table (now called Table 3), as some of the information contained therein could have been ascertained from the following figure.

Reviewer #3: The authors of this paper developed a deterministic transmission-dynamic model of HPV and HIV transmission, and natural history of HPV-related cervical cancer and HIV, to estimate the impact of actual and scaled-up HIV control actions on past and future trends of cervical cancer incidence and mortality in Tanzania. HIV control actions include voluntary medical male circumcision (VMMC), anti-retroviral therapy (ART), and pre-exposure prophylaxis. The model parameters were fixed based on the literature or calibrated to empirical data. Five scenarios of HIV control actions (no control, only VMMC, actual control with VMMC and ART, and three scaled-up control scenarios) were elaborated and compared in terms of cervical cancer incidence and mortality between 1995 and 2070. Age-standardized incidence of cervical cancer and mortality due to cervical cancer, number of deaths due to cervical cancers, number of cases of cervical cancer, and total number of death averted, HPV prevalence, HIV prevalence, were used as outcomes for comparison.

This study shows that interventions aimed at HIV prevention can have major indirect effect on cervical cancer. This important effect would be missed in a typical effectiveness analysis which does not include HPV infection and transmission. The authors’ usage of a model of HPV/HIV transmission and development of subsequent cervical cancer is necessary if HIV infection can affect both HPV transmission and HPV natural history of disease. The mathematical model used appears adequate overall with a potential limitation regarding the grouping of HPV-genotypes (e.g., a single group for all cross-protective types). However, the calibration does not account sufficiently for the substantial parameter uncertainty related to the complicated and still uncertain interaction between HIV and HPV, and such uncertainty is not visible in the results presented. Hence, the present study requires some methodological adjustment but is a potentially important contribution to the assessment of HIV prevention effectiveness in Tanzania and other sub-Saharan countries of similar context.

Major comments

The estimation of the effect of parameters uncertainty on the results is unclear or inadequate in this study. To my understanding, two sensitivity analyses were done. In the first analysis, a subset of the parameters is varied and their impact on calibration targets is assessed: this cannot be used to estimate parameters uncertainty on the study outcomes (HIV control actions effectiveness). Even if varying a parameter does not have an impact on calibration target this does not mean that the parameter does not have an impact on effectiveness. For example, different combinations of parameter can lead to the same fit of pre intervention incidence, but produce sharp differences in post intervention effectiveness. In the second analysis, parameters related to the effect of HIV control actions (e.g., effect of circumcision on transmission) are varied and their impact on the study outcomes are ascertained, but this analysis does not include other uncertain parameters such as those related to sexual behavior. In the presentation of results, a single number is given with no uncertainty interval. It is highly unlikely that the numbers given are precise enough to justify omitting an uncertainty interval.

We thank the reviewer for this comment and have attempted to address it in the following way. We have combined the two separate sensitivity analyses into one larger sensitivity analysis, which incorporates sexual behaviour (volume of interactions for high/general activity men and women, including over-time changes and the degree of age-assortative mixing), HIV and HPV transmission (sex-specific), the impact of HIV on HPV, the impact of VMMC on HIV and HPV and the effectiveness of PrEP. This sensitivity analysis was run for all modelled scenarios (see table 5).

To present the results of this sensitivity analysis in a concise but meaningful way for the reader, figure 11 depicts the possible variation for all modelled outcomes for all scenarios in the year 2070. 

The methodology of this modelling study is complex and is difficult to explain thoroughly in a short article. However, some essential information is missing and was moved to the supplementary materials: 1) information on calibration methods, 2) calibration data related to sexual behavior. Overall, although a lot of information on the methods are in Appendices, there should still be sufficient information in the core paper for readers to understand the approach and key simplifying assumptions.

We have taken this feedback on board and have moved substantial portions of the supplementary appendices to the main text (see responses above). In addition, the two separate supplementary appendices have been combined. Note that there still exists a supplementary appendix, which is now reserved for very large tables equations only. 

To help understand the results, they should include the overall impact of HIV interventions partitioned as the effect due to HIV reduction only plus the secondary effect on cervical cancer. This should be done with absolute and relative measures (number of deaths and mortality rates). Presenting only the absolute number of deaths averted is insufficient. Absolute numbers can be diluted by other effects such as population growth. Presenting both relative and absolute effects allow both to understand the underlying results (relative impact) and provide numbers for policy makers (absolute). In addition, relative numbers may help with generalizability (or discussion of generalizability) of the results to other settings.

We are unable to present a breakdown of the impact due to HIV reduction only in this iteration of the model design as the HIV and HPV components are fully integrated. 

To correct/account for the impact of population growth, we also present age-standardised rates (standardised to a fixed population structure and not changing over time) of cervical cancer incidence and mortality (figure 10). 

The limits related to grouping HPV-genotypes should be at the minimum mentioned. Grouping HPV-types is known to produce “super-bug” in the general population. In the context of this study with an important HIV-infected population, it is not immediately clear what would be the impact of grouping genotypes. Given that people living with HIV have a high prevalence of HPV co-infections, grouping types may have an impact on results as individuals in the model cannot be co-infected by more than 3 super types (what happens with competing risks towards HPV-related disease, transmission dynamics, etc?).

Thank you for pointing this out, as it is a limitation of the model. Please see our discussion on this point which has been inserted on page 35 (lines 450-455)

“The grouping of many individual HPV genotypes in the model, for example HPV 16/18, HPV 31/33/45/52/58 (HPV H5) and a category for “other high-risk HPV types”, may impact the overall simulated HPV prevalence in addition to the overall transmission dynamics of the model; for example, in this model it is impossible to discern whether individuals are infected with only one or any combination of the HPV genotypes in each simulated HPV subgroup. This may result in an overestimation of effectiveness of interventions targeted at HPV reduction”

I think more explanation of the following conclusions is needed: “These results demonstrate that HIV control measures will have a substantial effect in reducing HIV-related death in Tanzania, but will have an unintended consequence of increasing cervical cancer incidence and mortality in HIV-positive women for the next few decades, as a result of their increased life expectancy.” “These findings demonstrate the importance and urgency of scaling up cervical cancer prevention programs, such as HPV vaccination and cervical screening, as well as HIV control, in order to avoid the situation that lives saved from HIV-related death are instead lost to cervical cancer.”

We have clarified the concluding remarks by replacing the above with the following statement (line 469-479 p32-33)”

“While VMMC and ART can reduce the burden of cervical cancer in Tanzania in the long term, they are not sufficient to bring cervical cancer incidence beneath the threshold proposed by the WHO for cervical cancer elimination. Our finding that even under the best-case scenario the rate of cervical cancer incidence in all Tanzanian women is not reduced below 35 cases per 100,000 women per year (more than eight fold higher than the elimination threshold) demonstrates the importance and urgency of scaling up cervical cancer prevention programs, such as HPV vaccination and cervical screening, as well as HIV control, in order to avoid the situation that lives saved from HIV-related death are instead lost to cervical cancer. The WHO call for global action to eliminate cervical cancer as a public health problem is an important opportunity to galvanise and unite efforts to prevent cervical cancer in Tanzania and globally(64).”

In the results, the age-standardized mortality due to cervical cancer decreases after HIV prevention. Hence, it could be argued that HIV prevention has a positive effect on mortality rates due to cervical cancer. The increase is in absolute number of deaths due to cervical cancer, because deaths have been prevented and people are living longer. It would be good to have more discussion on this, so that readers better understand the implications. If childhood mortality is prevented, there will be more deaths among adults. If deaths are prevented among young adults, more deaths will occur among older adults, etc. Lives saved from HIV will results in substantial life years gained, which should not be mistakenly interpreted as something negative. Rather that it will have consequences on the number of cervical cancer cases, and thus provide even more justification for cervical cancer prevention. But not only for cervical cancer prevention as it will increase the number of deaths due to other prevalent diseases among people living with HIV.

We now explicitly address this point in the discussion (p 32 lines 456-457): “Finally, the findings of this study must be interpreted in the context of the lengthening life expectancy in Tanzania. With the life expectancy at birth expected to rise to 75 years in 2065-2070 (compared to 54 years in 1995-2000)(34) this will necessarily result in an increased opportunity for the development of cervical cancer (and other diseases), irrespective of additional effects due to HIV treatment.”

 

Minor comments

In the No intervention scenario, condom usage declines between 2011 and 2016. However, could this be partially due to the HIV prevention measures? The authors mention that HIV prevention can affect sexual behavior and condom usage. If so, shouldn’t the decline in condom usage be ignored in the No intervention scenario?

This is an interesting point, however, in this case we believe that it would be most prudent to simulate the observed data in this case. Since the objective of this paper is to investigate the effects of VMMC and ART, modifying the underlying condom usage assumptions for one scenario would confuse the effect of the target interventions. While we do include some discussion and incorporation of behavioural epidemiology in this article, it is not the main focus of this study. 

In First Supplementary Appendix to the article:

P. 3, last paragraph: "The number of high-risk sexual interactions is defined as the total number of sexual interactions which could potentially result in HIV transmission." This sentence needs to be re-written, unless the assumption is that low-risk sexual interaction can’t result in HIV transmission.

Please note that (as per the above) the sexual behaviour and force of infection section of this manuscript has been completely re-written (and relocated to the main text). The original statement quoted above is worded incorrectly, as it is true that low-risk sexual interactions can still result in HIV transmission. Please note that (in Table 5 of the main text), we now refer to “the volume of sexual interactions possibly resulting in HIV transmission …”.

---

## [Decision Letter · Decision Letter 1]

24 Mar 2020

The past, present and future impact of HIV prevention and control on HPV and cervical disease in Tanzania: a modelling study.

PONE-D-19-24871R1

Dear Dr. Hall,

We are pleased to inform you that your manuscript has been judged scientifically suitable for publication and will be formally accepted for publication once it complies with all outstanding technical requirements.

With kind regards,

Alberto d'Onofrio, Ph.D.

Academic Editor

PLOS ONE

Additional Editor Comments (optional):

Dear Authors,

based on

1)a positive full referee report;

2) on the positive comments received by a second referee that could, however, not to send a formal assessment of your revised ms

and

3) on my personal reading of your very good work,

I am happy to communicate to you that your revised manuscript is now accepted.

We apologize for the delays, due in part to the complexity of your work and in part to the current Covid-1 emergency on which

many referees are working on (and also myself)

I am sure that you perfectly understand and will be so kind to forgive us this delay.

With Best Regards,

Alberto d'Onofrio

Reviewers' comments:

Reviewer's Responses to Questions

**Comments to the Author**

1. If the authors have adequately addressed your comments raised in a previous round of review and you feel that this manuscript is now acceptable for publication, you may indicate that here to bypass the “Comments to the Author” section, enter your conflict of interest statement in the “Confidential to Editor” section, and submit your "Accept" recommendation.

Reviewer #2: All comments have been addressed

2. Is the manuscript technically sound, and do the data support the conclusions?

Reviewer #2: Yes

3. Has the statistical analysis been performed appropriately and rigorously? 

Reviewer #2: Yes

4. Have the authors made all data underlying the findings in their manuscript fully available?

Reviewer #2: (No Response)

5. Is the manuscript presented in an intelligible fashion and written in standard English?

Reviewer #2: Yes

6. Review Comments to the Author

Reviewer #2: (No Response)

7. PLOS authors have the option to publish the peer review history of their article (what does this mean?). If published, this will include your full peer review and any attached files.

Reviewer #2: Yes: Piero Manfredi

---

## [Editor Report · Acceptance letter]

20 Apr 2020

PONE-D-19-24871R1 

The past, present and future impact of HIV prevention and control on HPV and cervical disease in Tanzania: a modelling study. 

Dear Dr. Hall:

I am pleased to inform you that your manuscript has been deemed suitable for publication in PLOS ONE. Congratulations! Your manuscript is now with our production department. 

With kind regards,

on behalf of

Dr. Alberto d'Onofrio 

Academic Editor

PLOS ONE